# Unifying Reconstruction and Density Estimation via Invertible Contraction Mapping in One-Class Classification

**Xiaolei Wang**[1,2]  **Tianhong Dai**[4]  **Huihui Bai**[3]  **Yao Zhao**[3]  **Jimin Xiao**[1*]

[1]Xi'an Jiaotong-Liverpool University
[2]University of Liverpool
[3]Beijing Jiaotong University
[4]Applied Computing Technologies

## Abstract

Due to the difficulty in collecting all unexpected abnormal patterns, One-Class Classification (OCC) has become the most popular approach to anomaly detection (AD). Reconstruction-based AD method relies on the discrepancy between inputs and reconstructed results to identify unobserved anomalies. However, recent methods trained only on normal samples may generalize to certain abnormal inputs, leading to well-reconstructed anomalies and degraded performance. To address this, we constrain reconstructions to remain on the normal manifold using a novel AD framework based on contraction mapping. This mapping guarantees that any input converges to a fixed point through iterations of this mapping. Based on this property, training the contraction mapping using only normal data ensures that its fixed point lies within the normal manifold. As a result, abnormal inputs are iteratively transformed toward the normal manifold, increasing the reconstruction error. In addition, the inherent invertibility of contraction mapping enables flow-based density estimation, where a prior distribution learned from the previous reconstruction is used to estimate the input likelihood for anomaly detection, further improving the performance. Using both mechanisms, we propose a bidirectional structure with forward reconstruction and backward density estimation. Extensive experiments on tabular data, natural image, and industrial image data demonstrate the effectiveness of our method. Code is available at URD.

## 1 Introduction

Anomaly detection (AD) focuses on identifying unexpected patterns that deviate from the known normal ones. Due to difficulty of collecting abnormal samples, One-Class Classification (OCC) paradigm [38, 10, 4, 34, 41, 43, 45, 18, 49] has become the primary solution for unsupervised anomaly detection (UAD) tasks. In the OCC paradigm, only normal data is provided during training, and the model is expected to detect previously unseen anomalies during inference. Therefore, the key challenge of the OCC paradigm involves in effectively modeling the normality of the training data to recognize the abnormal pattern. Recently, reconstruction-based [12, 14, 54, 19, 30, 21, 48] and density-estimation-based [39, 26, 55, 10, 58] methods are the two most widely used OCC AD methods. The former relies on the assumption that a model trained solely on normal data will be unable to accurately reconstruct anomalies. In contrast, the latter focuses on fitting a probability distribution over normal data to estimate the likelihood of previously unseen inputs.

---

[*]Corresponding Author

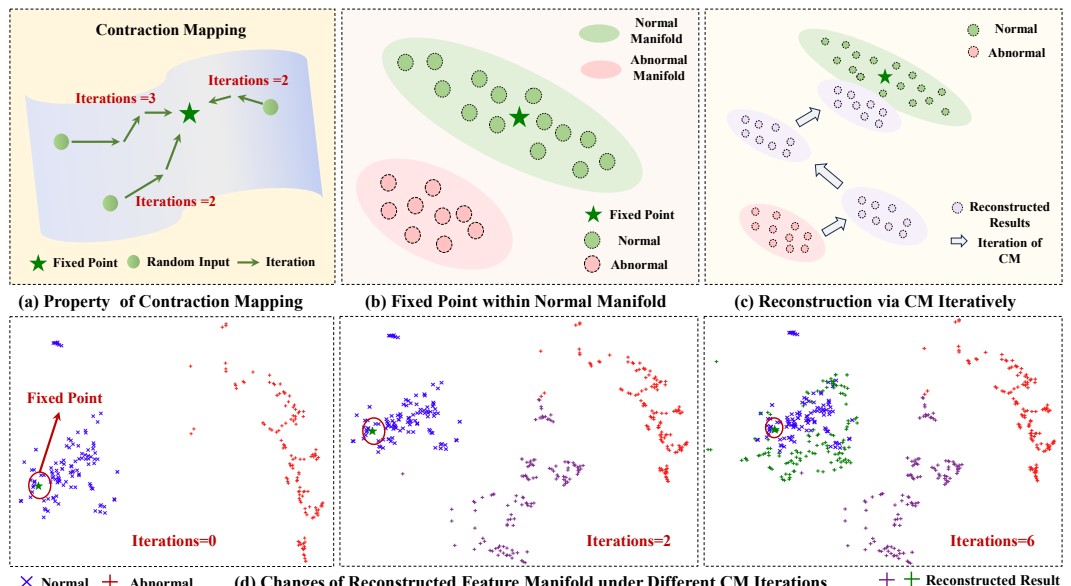

Figure 1: (a): Any input converges to a fixed point after several CM iterations. (b): The fixed point of CM trained by normal data is located within the normal manifold. (c): Abnormal inputs (red) are reconstructed into the normal manifold (green) by several iterations of CM. (d) T-SNE visualization of changes in the reconstructed feature manifold on the 'Ionosphere' tabular AD dataset [36].

Specifically, reconstruction-based methods rely on the discrepancy between the input and its reconstructed output to identify unseen anomalies. However, models trained exclusively on normal data may generalize to certain abnormal inputs, resulting in well-reconstructed anomalies and reduced reconstruction error, which affects the performance. To address this issue, previous AD methods [54, 53] adopt reconstruction strategies based on partially masked data. However, when using handcrafted or random mask strategies, not all abnormal information can be covered, so they still cannot prevent well-reconstructed anomalies and guarantee the effectiveness of all obtained reconstruction discrepancies. Thus, the key question remains: *How can reconstruction discrepancies be made sufficiently discriminative for detecting each abnormal input?*

An intuitive way to obtain discriminative reconstruction discrepancy is to constrain reconstructed results of abnormal inputs remain on the normal manifold. To this end, we propose a novel reconstruction network based on contraction mapping (CM). Based on the property of CM [16], any input is theoretically guaranteed to converge to a unique fixed point through iterative applications of a CM (see Fig.1 (a)), and this point depends exclusively on the mapping itself. Inspired by the property of CM, we further observe during training that optimizing our CM-based network with a reconstruction loss on normal data encourages the fixed point to lie within the normal manifold (see Fig.1 (b)). Therefore, given abnormal inputs, we can leverage trained CM iteratively to transform them into the normal manifold (see Fig.1 (c)). T-SNE visualization of real data in Fig.1 (d) also shows that the reconstructed results are constrained to remain on the normal manifold after six iterations of CM, resulting in a significant discrepancy between the abnormal input and its reconstruction.

On the other hand, our proposed CM structure is inherently invertible. According to the theory of normalizing flow [13, 27], an invertible mapping flow can transform a Gaussian prior into a real data distribution, which enables the estimation of the likelihood of the input. Therefore, our invertible CM flow is leveraged to perform density estimation to further improve the detection performance. In particular, after the reconstruction stage, the reconstructed output is used to learn a prior Gaussian distribution. Then, we leverage the inverse of the contraction mapping flow on the prior distribution to estimate the likelihood of the inputs. To this end, our framework consists of forward reconstruction and backward density estimation, unifying these two popular paradigms to detect unseen anomalies. The main contributions of this paper are summarized as follows:

**(1)** To prevent well-reconstructed anomalies, we adopt a novel contraction mapping structure to constrain abnormal inputs onto the normal manifold. **(2)** A unified framework is introduced that

combines reconstruction and density estimation using an invertible contraction mapping flow, which bridges two popular paradigms in anomaly detection. **(3)** Extensive experiments on tabular, natural image, and industrial image data demonstrate the effectiveness of our method.

## 2 Related Work

**Anomaly Detection**  Anomaly detection aims to recognize unexpected abnormal patterns in different types of data, such as tabular [53, 9], graph [57], and image data [47, 8, 44]. Since the form and type of anomaly events are unpredictable and cannot be fully collected, One-Class-Classification-based AD paradigm has become the mainstream solution. The key of OCC approaches is to model the normality of training data to detect unseen anomalies. Classic OCC methods [35, 6, 40, 46, 38, 57] learn a profile from normal data, computing the distance between the abnormal input and the center/boundary of the profile as the anomaly score. KNN [35] and LOF [6] directly compute the distance between a test input and its several normal neighbors. OCSVM [40] learn a linear boundary of normal samples to discriminate anomalies. SVDD-based approaches [46, 38, 57] construct hypersphere boundaries to effectively surround normal representations, recognizing abnormal patterns based on the distance between these representations and the center/boundary of the sphere.

**Reconstruction-based Method**  Reconstruction-based methods rely on the discrepancy of inputs and their reconstructed results to identify unseen anomalies. However, a trained reconstruction network may recover some anomalies well, obtaining a small reconstruction error and failing to detect anomalies. To alleviate this issue, the tabular AD method, MCM[53] adopts a learnable mask to cover partial areas of the original input and learning normal reconstruction from masked data. UniAD [54], a classic visual AD method, defines this issue as 'identical shortcut' and similarly adopts the neighbor mask in attention computation to alleviate well-reconstructed anomalies. The graph-based AD method [22] leverages the denoising autoencoder to learn the reconstruction for the noisy perturbation. In this paper, we consider that as long as the reconstructed output can be constrained within the normal manifold, well-reconstructed anomalies will be avoided. Based on the above idea, we propose to apply contraction mapping to conduct reconstruction.

**Density-Estimation-based Method**  Density-estimation-based methods [50, 58] apply normal data to fit a Gaussian distribution or its variants to estimate the likelihood of each test input as the anomaly score. However, the real normal data distribution is much more complex than a Gaussian distribution. Therefore, normalizing-flow-based AD methods [51, 39, 55, 17] adopt invertible mapping to transform a prior distribution of latent variables to the distribution of real inputs, better capturing the probability density of each input. In this paper, the introduced contraction mapping flow is designed to be invertible and can directly estimate the likelihood of input based on a learnable Gaussian prior distribution of reconstructed results. Our framework consists of forward reconstruction and backward density estimation, unifying these two main paradigms.

## 3 Methodology

**Preliminary**  In the one-class classification (OCC) anomaly detection (AD) task, only normal samples are provided during training, and the goal is to detect previously unseen anomalies in the testing set. Formally, $\mathcal{I} = \mathcal{I}^{\text{train}} \cup \mathcal{I}^{\text{test}}$ is represented as an AD dataset, where the data can be either tabular or image-based. The training set is denoted as $\mathcal{I}^{\text{train}} = \{I_i, y_i\}_{i=1}^{N_1}$, where each $I_i$ is a normal sample with label $y_i = 0$, and $N_1$ is the number of samples. The testing set is denoted as $\mathcal{I}^{\text{test}} = \{I_i^t, y_i^t\}_{i=1}^{N_2}$, which contains both normal and abnormal samples, $y_i^t \in \{0, 1\}$, and $N_2$ is the number of testing samples.

**Overview**  As shown in Fig.2, our proposed framework consists of two main components: 1) forward reconstruction and 2) backward density estimation. This unified design integrates both reconstruction-based and density-based anomaly detection within a single framework, resulting in enhanced detection performance. During training, the normal input $I$ is first passed through a standard neural network to achieve the corresponding representations $\mathbf{x} \in \mathbb{R}^d$, where $d$ is the dimension of the vector. Next, to mitigate the issues of 'reconstructing anomalies well', a novel contraction mapping (CM) based network $\varphi(\cdot)$ is proposed. This network is trained with a reconstruction loss on

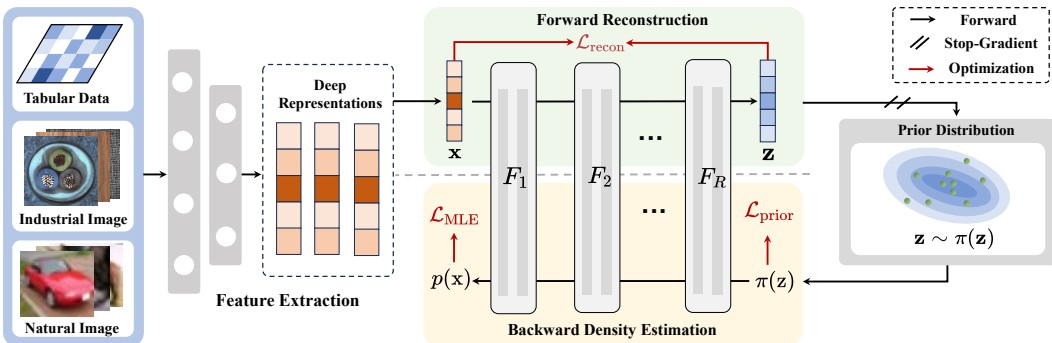

Figure 2: Overview of our proposed method. For a normal input, we first extract its deep representation. Next, we apply contraction mapping flow to reconstruct the obtained representation. Then, reconstructed outputs are used to fit a prior distribution. Finally, the reverse of the mapping flow can transform this prior to the input distribution, enabling likelihood estimation of the original input.

normal samples to constrain reconstructed results to lie within the normal manifold. Specifically, the contraction mapping is used in a flow-based manner to iteratively constrain the reconstruction within the normal manifold, and the final reconstructed result is denoted as $\mathbf{z} \in \mathbb{R}^d$. To further exploit the inherently invertibility of CM, we leverage the reverse mapping $\varphi^{-1}(\cdot)$ to estimate the density $p(\mathbf{x})$ of the input $\mathbf{x}$ from a learnable prior distribution over the latent variable $\mathbf{z}$. Then, the flow-based reverse mapping $\varphi^{-1}(\cdot)$ gradually transforms the prior Gaussian distribution $\pi(\cdot)$ to the density $p(\cdot)$, which captures the likelihood of the real input $\mathbf{x}$. During inference, a combination of reconstruction error and likelihood is adopted as the ultimate anomaly score.

## 3.1 Reconstruction via Contraction Mapping

Anomaly detection via reconstruction relies on the discrepancy between the input and its reconstruction. Typically, reconstruction-based methods employ the following objective function to train the reconstruction network $\varphi(\cdot)$:

$$\mathcal{L}_{\text{rec}} = \|\mathbf{z} - \mathbf{x}\|_2, \quad \mathbf{z} = \varphi(\mathbf{x}), \tag{1}$$

where $\mathbf{x}$ is the input feature, $\mathbf{z}$ is the corresponding reconstruction, and $\| \cdot \|_2$ is $\ell_2$-norm. During inference, the reconstruction error $\|\varphi(\mathbf{x}^t) - \mathbf{x}^t\|_2$ between the input feature x and its reconstruction is adopted as the anomaly score for the test data. However, as discussed in Section 1, recent methods often generalize to anomaly inputs, producing low reconstruction error and thus failing to identify anomalies efficiently. To address this issue, we model $\varphi(\cdot)$ as a contraction mapping with a flow-based reconstruction process, enforcing that the reconstructed output from any input remains on the normal manifold. This leads to larger reconstruction errors when receiving anomaly inputs.

### 3.1.1 Theoretical Analysis of Contraction Mapping Reconstruction

In this section, we provide formal mathematical proofs to demonstrate how the contraction mapping (CM) constrains the reconstruction to remain on the normal manifold. Specifically, we prove two key propositions: 1) training the proposed CM-based network with a reconstruction loss on normal data ensures that the fixed point of the CM lies within the normal manifold, and 2) the trained CM network can transform the reconstruction of abnormal inputs toward the normal manifold.

First, we introduce the formal definition and the property of the contraction mapping.

**Definition 1** *(Contraction Mapping [25]): A mapping $f : \mathbb{R}^d \longrightarrow \mathbb{R}^d$ is called a contraction mapping if there exists a constant $\eta \in [0, 1)$ such that $\|f(x_1) - f(x_2)\|_2 \leq \eta \|x_1 - x_2\|_2$ for all $x_1, x_2 \in \mathbb{R}^d$. The constant $\eta$ is called the Lipschitz constant.*

**Lemma 1** *(Convergence of Iterations [25]): Let $f(\cdot)$ be a contraction mapping with $\eta \in [0, 1)$. For any initial input $x_0 \in \mathbb{R}^d$, we define the sequence $\{x_n\}$ by $x_{n+1} = f(x_n)$ for $n = 1, 2, \cdots$. Then $\lim_{n \to \infty} x_n = x^*$, where $x^*$ is unique and is called a fixed point of $f(\cdot)$.*

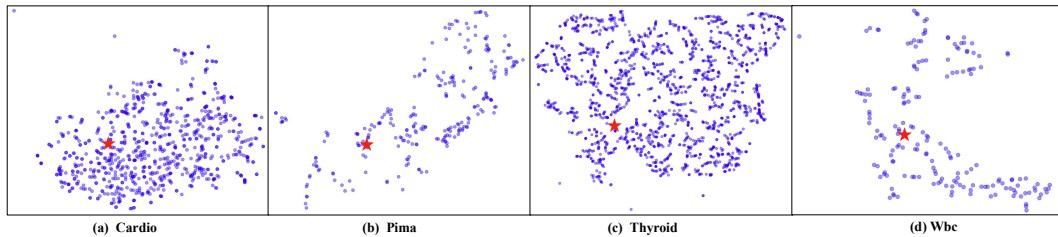

(a) Cardio       (b) Pima       (c) Thyroid       (d) Wbc

Figure 3: T-SNE visualization of the fixed point on four AD datasets. Training $f(\cdot)$ with $\mathcal{L}_{\text{rec}}$ on normal data ensures that the fixed point of $f(\cdot)$ lies within the normal manifold.

Lemma 1 shows that any input will converge to a fixed point after iterative application of a contraction mapping (see Fig.1 (a)), where the point is only related to that mapping itself. Next, the contraction mapping is modeled by a learnable network $f^{\theta}(\cdot)$, where $\theta$ denotes the parameters. To simplify the notation, $\theta$ is omitted. Empirically, we observe that training $f(\cdot)$ on normal data using reconstruction loss $\mathcal{L}_{\text{rec}} = \|f(\mathbf{x}) - \mathbf{x}\|_2$ result in a fixed point that lies within the normal manifold. To theoretically support this observation, we propose the following proposition:

**Proposition 1** *Let $f(\cdot)$ be a contraction mapping with $\eta \in [0, 1)$, and let $\mathbf{x} \in \mathbb{R}^d$ be a normal sample. Under the optimization of the reconstruction loss $\mathcal{L}_{\text{rec}} = \|f(\mathbf{x}) - \mathbf{x}\|_2$, the fixed point $\mathbf{x}^* \in \mathbb{R}^d$ of $f(\cdot)$ is located within a small neighborhood of the normal input $\mathbf{x}$, and $\|\mathbf{x}^* - \mathbf{x}\|_2 \leq \frac{\mathcal{L}_{\text{rec}}}{1-\eta}$.*

*proof. Let us assume that $f(\cdot)$ is a contraction mapping iteration with $L \in [0, 1)$. According to Definition 1 and Lemma 1, one inequality can be obtained: $\|\mathbf{x}^* - f(\mathbf{x})\|_2 = \|f(\mathbf{x}^*) - f(\mathbf{x})\|_2 \leq \eta\|\mathbf{x}^* - \mathbf{x}\|_2$. Next, using the triangle inequality, we obtain $\|\mathbf{x}^* - \mathbf{x}\|_2 = \|\mathbf{x}^* - f(\mathbf{x}) + f(\mathbf{x}) - \mathbf{x}\|_2 \leq \|\mathbf{x}^* - f(\mathbf{x})\|_2 + \|f(\mathbf{x}) - \mathbf{x}\|_2$. Combining the above two inequalities, one can be obtained: $\|\mathbf{x}^* - \mathbf{x}\|_2 \leq \frac{1}{1-\eta}\|f(\mathbf{x}) - \mathbf{x}\|_2 = \frac{\mathcal{L}_{\text{rec}}}{1-\eta}$. When $\mathcal{L}_{\text{rec}}$ is minimized during training, the fixed point $\mathbf{x}^*$ must be in the vicinity of the normal input $\mathbf{x}$.*

Proposition 1 provides a theoretical analysis of the reconstruction for a single sample. In practice, during training, all normal samples are used to optimize the network $f(\cdot)$, which drives the fixed point $\mathbf{x}^*$ to converge to the interior of the normal manifold. The empirical visualizations in Fig. 3 on real datasets strongly support this theoretical claim, showing the fixed point landing within the normal data manifold.

Next, based on Proposition 1 and Lemma 1, any input can be iteratively mapped to a fixed point within the normal manifold by applying $f(\cdot)$ recursively. Therefore, inspired by normalizing flow [13], we adopt a flow-based module $\varphi_K(\cdot) = f_K \circ f_{K-1} \circ \cdots f_1(\cdot)$ to model the iterative process of the contraction mapping, where each $f_k(\cdot)$ shares the same structure but has its own parameters $\theta_k$. Given a test abnormal input $\mathbf{x}^t$, the flow-based module $\varphi_K(\cdot)$ can transform it toward a fixed point $\mathbf{x}^*$ within the normal manifold, thus inducing a large reconstruction error $\|\varphi_K(\mathbf{x}^t) - \mathbf{x}^t\|_2$. To theoretically support this perspective, we introduce a key proposition along with a related lemma:

**Lemma 2** *Let $\varphi_K(\cdot) = f_K \circ f_{K-1} \circ \cdots f_1(\cdot)$, where each $f_k$ is contraction mapping with $\eta_k \in [0, 1)$, then $\varphi_K(\cdot)$ is a contraction mapping with Lipschitz constant $\eta_{\varphi} \leq \eta_1 \cdot \eta_2 \cdots \eta_K$.*

**Proposition 2** *Let $\varphi_K(\cdot) = f_K \circ f_{K-1} \circ \cdots f_1(\cdot)$ is a contraction mapping flow trained by reconstruction loss $\mathcal{L}_{\text{rec}} = \|\varphi_K(\mathbf{x}) - \mathbf{x}\|_2$, where $\mathbf{x}$ is normal. The fixed point $\mathbf{x}^*$ of $\varphi_K(\cdot)$ is located within a small neighborhood of $\mathbf{x}$. Additionally, for any abnormal input $\mathbf{x}^t$, as the value of $K$ increases, $\varphi_K(\mathbf{x}^t)$ is closer to the fixed point $\mathbf{x}^*$, thereby being closer to normal objective $\mathbf{x}$.*

*proof. According to Lemma 2, $\varphi_K(\cdot)$ is a contraction mapping with Lipschitz constant $\eta_{\varphi}$, so it also exists a unique fixed point $\mathbf{x}^*$. According to Proposition 1, under optimization of $\mathcal{L}_{\text{rec}}$, one inequality can be obtained: $\|\mathbf{x}^* - \mathbf{x}\|_2 \leq \frac{\mathcal{L}_{\text{rec}}}{1-\eta_{\varphi}}$, where $\eta_{\varphi} \leq \eta_1 \cdot \eta_2 \cdots \eta_K < 1$. Therefore, as minimizing $\mathcal{L}_{\text{rec}}$ the fixed point $\mathbf{x}^*$ must be in the vicinity of normal $\mathbf{x}$. Next, for an abnormal input $\mathbf{x}^t$, the distance between its reconstructed result $\varphi_K(\mathbf{x}^t)$ and the fixed point $\mathbf{x}^*$ is: $\|\varphi_K(\mathbf{x}^t) - \mathbf{x}^*\|_2 \leq \eta_{\varphi}\|\mathbf{x}^t - \mathbf{x}^*\|_2$. Therefore, as $K$ increases, $\eta_{\varphi}$ decreases and $\varphi_K(\mathbf{x}^t)$ is closer to the fixed point. Moreover, according to triangle inequality and above inequality, one can be obtained: $\|\varphi_K(\mathbf{x}^t) - \mathbf{x}\|_2 \leq \|\varphi_K(\mathbf{x}^t) - \mathbf{x}^*\|_2 + \|\mathbf{x}^* - \mathbf{x}\|_2 \leq \eta_{\varphi}\|\mathbf{x}^t - \mathbf{x}^*\|_2 + \frac{\mathcal{L}_{\text{rec}}}{1-\eta_{\varphi}}$. As $K$ increases, $\eta_{\varphi}$ decreases, reducing the first term. The second term also decreases due to $1/(1 - \eta_{\varphi})$ where $\mathcal{L}_{\text{rec}}$ is a small constant through training.*

With the support of Proposition 2, any abnormal input can be transformed into its corresponding normal representation by $\varphi_K(\cdot)$, resulting in more discriminative reconstruction errors and improved detection performance. Fig.1 (d) shows a T-SNE visualization of the iterative mapping results of $\varphi_K(\cdot)$ for different values of $K$. Proofs of Lemma 1 and 2 are provided in the appendix.

Since the mapping $\varphi_K(\cdot)$ is trained for reconstruction by minimizing the loss on normal samples, it effectively behaves as an approximate identity mapping for normal inputs. It also retains its contraction property, pulling abnormal inputs toward the fixed point, which is clearly illustrated in Fig.4. Moreover, in practice, $K$ is kept small to prevent all inputs from converging to the fixed point.

### 3.1.2 Modeling of Contraction Mapping Iteration

In this section, based on the above theoretical analysis, a detailed description of the structure of the flow-based contraction mapping $\varphi(\cdot)$ is provided. First, inspired by [3, 1], each contraction mapping unit is represented as $f(\cdot)$:

$$f(x) = \left(\frac{\mathbf{Id} + \mathbf{G}}{2}\right)^{-1}(x) - x, \tag{2}$$

where $\mathbf{G}$ is designed as a Lipschitz-continuous invertible mapping with Lipschitz constant $\eta < 1$. For practical implementation, a block-wise approach is used to build $\varphi(\cdot)$. Specifically, a contraction mapping flow module $F$ is constructed as a composition of $K$ identical mapping units, i.e., $F(\cdot) = f_K \circ f_{K-1} \circ \cdots f_1(\cdot)$. Finally, the contraction mapping flow $\varphi(\cdot)$ can be written as:

$$\varphi(\cdot) = F_R \circ F_{R-1} \cdots F_1(\cdot), \tag{3}$$

where $R$ is the number of flow modules. As a reconstruction network, the primary objective is to learn a contraction mapping flow $\varphi(\cdot)$ via Eq.(1), which enables the reconstruction of abnormal inputs toward the normal manifold without forcing full convergence to the final fixed point. Thus, $K$ and $R$ are kept relatively small to avoid full convergence and maintain the detection performance.

In addition, to implement $\varphi(\cdot)$, i-DenseNet [33] is applied to parametrize the mapping $\mathbf{G}$. The basic layer of i-DenseNet can be written as $\psi(\mathbf{W}x + \mathbf{b})$, where weight $\|\mathbf{W}\|_2 < 1$, $\|\cdot\|_2$ denotes the spectral norm ($\mathbf{W}$ is a matrix), and $\psi$ is 1-Lipschitz activation function. The above constraints ensure that the Lipschitz constant of $\mathbf{G}$ is less than 1. More details about the implementation of invertible neural network $\varphi(\cdot)$ are provided in the appendix.

## 3.2 Density Estimation via Reversed Flow

Based on Eq.(2), the contraction mapping $f(\cdot)$ is an invertible mapping and $f^{-1}(x) = ((\mathbf{Id} - \mathbf{G})/2)^{-1}(x) - x$ (the proof can be found in [1]). Therefore, the flow module $\varphi(\cdot)$ is also invertible and $\varphi^{-1}(\cdot)$ can be written as $\varphi^{-1}(\cdot) = F_1^{-1} \circ F_2^{-1} \cdots F_R^{-1}(\cdot)$.

As established in the framework of normalizing flows [13, 27], $\varphi^{-1}(\cdot)$ is able to transform a Gaussian prior distribution into a real data distribution, allowing for estimating the likelihood of the input. In this section, the previous reconstructed output $\mathbf{z}$ is used to learn a Gaussian prior distribution $\pi(\mathbf{z})$. Then, the reversed flow $\varphi^{-1}(\cdot)$ is applied on the prior distribution $\pi(\mathbf{z})$ to achieve the log-likelihood $\log p(\mathbf{x})$ of the input $\mathbf{x}$:

$$\log p(\mathbf{x}) = \log \pi(\mathbf{z}) - \sum_{r=1}^{R} \log |\det(\mathbf{J}_{F_r^{-1}}(\mathbf{z}_{r-1})|, \tag{4}$$

where $|\det(\cdot)|$ is the magnitude of the determinant and $\mathbf{J}_{F_r^{-1}}$ is the Jacobian matrix of each $F_r^{-1}$, each $\mathbf{z}_r$ is the output of the module $F_r^{-1}$, i.e., $\mathbf{z}_{r-1} = F_r^{-1}(\mathbf{z}_r)$, and $\mathbf{z}_R = \mathbf{z}$. Next, we will show more details of prior distribution $\pi(\cdot)$ and optimization objective.

**Prior Distribution** $\pi(\cdot)$   According to Eq.(4), we need to compute $\log \pi(\mathbf{z})$ for the final $\log p(\mathbf{x})$. Given a reconstructed output $\mathbf{z}$, the learnable parameters $\mu$ and $\Sigma$ of the prior Gaussian distribution $\pi(\cdot)$ are updated through maximizing $\log \pi(\mathbf{z})$, where $\mathbf{z} \sim \pi(\mathbf{z})$. A stop-gradient operation $\mathrm{sg}(\cdot)$ is employed during the optimization of prior distribution $\pi(\cdot)$, which allows it to be learned independently without affecting the reconstruction training:

$$\mathcal{L}_{\mathrm{prior}} = \mathbb{E}[-\log \pi(\mathrm{sg}(\mathbf{z}))], \tag{5}$$

where $\mathbb{E}[\cdot]$ represents the mathematical expectation operator. Next, the distribution $p(\mathbf{x})$ is estimated by transforming the prior distribution $\pi(\mathbf{z})$ through $\varphi^{-1}(\cdot)$.

**Optimization Objective**    Based on normalizing flows [13], we adopt maximizing log likelihood to train flow $\varphi(\cdot)$. According to Eq.(4), $\mathcal{L}_{\text{MLE}}$ implicitly optimizes both the prior distribution $\pi(\cdot)$ and the parameters of our INN $\varphi(\cdot)$, i.e.,

$$\mathcal{L}_{\text{MLE}} = \mathbb{E}[-\log p(\mathbf{x})] = \mathbb{E}[-\log \pi(\text{sg}(\mathbf{z}))] + \mathbb{E}\Big[\sum_{r=1}^{R} \log |\det(\mathbf{J}_{F_r^{-1}}(\mathbf{z}_{r-1})|\Big] \quad (6)$$

Combining Eq.(1) and (6), the ultimate loss function is defined as:

$$\mathcal{L}_{\text{total}} = \mathcal{L}_{\text{rec}} + \lambda \mathcal{L}_{\text{MLE}}, \quad (7)$$

where $\lambda$ is a hyper-parameter. The entire framework is trained in an end-to-end manner.

### 3.3  Inference and Anomaly Score

During inference, the feature vector $\mathbf{x}^t$ of a given input $I^t$ is obtained through the feature extraction network. Then, in the forward reconstruction, the reconstruction $\mathbf{z}^t$ is achieved by using the contraction mapping flow $\varphi(\cdot)$ to get the reconstruction error. Next, the log-likelihood $\log p(\mathbf{x}^t)$ of the input is computed according to Eq.(4) and $\log \pi(\mathbf{x})$. Therefore, the final anomaly score is represented as:

$$S(\mathbf{x}^t) = \|\varphi(\mathbf{x}^t) - \mathbf{x}^t\|_2 - \alpha \log p(\mathbf{x}^t), \quad (8)$$

where $\alpha$ is a weight coefficient to control the scale of $\log p(\mathbf{x}^t)$.

## 4  Experiment

**Datasets**    In this paper, our method is evaluated on anomaly detection (AD) tasks across tabular, graph, and image domains. For tabular AD, we select 12 benchmark datasets from OODS [36] and ADBench [20], which include diverse fields, such as healthcare, finance, and social sciences, etc. For visual AD tasks, we conduct experiments on both natural and industrial datasets: CIFAR-10 and MVTec AD [5], respectively. Following previous OCC works [38, 57], only one class in CIFAR-10 is selected as the normal class, while the remaining classes are considered abnormal.

**Evaluation Metrics**    For tabular data, we adopt the Area Under the Receiver Operating Characteristic Curve (AUROC) and Area Under the Precision-Recall Curve (AP) as evaluation metrics. For image data, image-level AUROC (I-AUROC) is used for anomaly classification and pixel-level AUROC is used for anomaly localization.

**Implementation Details**    The implementation is based on PyTorch. In the tabular AD task, we use a three-layer Multilayer Perceptron (MLP) with LeakyReLU activation function as a feature extractor. The flow-based structure is modeled using $K = 2$ and $R = 3$. In particular, the hyperparameters are set to $\lambda = 0.5$ and $\alpha = 0.01$ across 20 tabular datasets. During training, Adam optimizer is used to update the network, where learning rate and weight decay are $1 \times 10^{-5}$ and $1 \times 10^{-7}$, respectively. For natural image datasets, CIFIAR-10, we use the same convolutional neural network (CNN) as DO2HSC [57] to extract the deep representation of each image. Same as the tabular setting, we set $K = 2$, $R = 3$, $\lambda = 0.5$, and $\alpha = 0.001$. The learning rate and weight decay are set to $2 \times 10^{-5}$ and $1 \times 10^{-7}$, respectively. We train the model for 50 epochs with a batch size of 32. For the industrial dataset MVTec, a frozen WideResNet-50 [23] is used as the backbone for both the encoder and decoder. For fair comparison, each image is resized to $224 \times 224$ without any data augmentation. We select $K = 3$, $R = 3$, $\lambda = 0.5$, and $\alpha = 0.001$ in the framework. During training, the learning rate and the weight decay are selected as $5 \times 10^{-4}$ and $1 \times 10^{-6}$, respectively. We train the model for 100 epochs with a batch size of 16. All experiments are conducted on a single NVIDIA RTX 3090 24GB GPU. More details can be found in the appendix.

### 4.1  Main Results

**Tabular OCC Task**    Tab.1 demonstrates that our framework outperforms previous state-of-the-art (SOTA) methods [29, 6, 40, 38, 58, 42, 53] and achieves a competitive performance with an AP of **78.64** and an AUROC of **91.19**. Compared to density-estimation-based methods [6, 38, 58], our framework adopts the property of the invertible mapping to compute the log-likelihood of each input

Table 1: Comparison of AP/AUROC results with the previous methods on 20 different tabular datasets. Following MCM [53], all datasets are implemented with identical dataset partitioning. 'DSVDD' stands for 'DeepSVDD'. The best result is highlighted in **bold**.

| Dataset | IForest[29] | LOF[6] | OCSVM[40] | DSVDD[38] | DGMM[58] | ICL[42] | MCM[53] | Ours |
|---|---|---|---|---|---|---|---|---|
| Arrhythmia | 50.97/77.34 | 52.77/76.88 | 53.39/76.89 | 60.36/79.40 | 46.68/72.83 | **61.55/81.45** | 61.07/81.14 | 59.63/77.56 |
| Breastw | 94.49/97.19 | 99.23/99.37 | 99.34/99.38 | 99.24/99.25 | 75.84/76.05 | 96.56/97.25 | **99.52/99.55** | 99.33/99.43 |
| Cardio | 70.18/92.20 | 83.60/95.62 | 86.14/**96.54** | 78.80/93.13 | 30.89/69.51 | 80.37/95.14 | 84.89/96.03 | **86.35**/96.21 |
| Census | 13.57/60.15 | 23.43/71.10 | 22.79/71.58 | 15.14/63.27 | 10.66/45.06 | 19.49/68.31 | 24.20/75.81 | **37.68/80.64** |
| Campaign | 46.08/72.69 | 44.59/70.61 | 47.49/76.26 | 25.29/52.54 | 24.72/57.85 | 45.06/72.41 | **60.40/89.02** | 55.31/84.49 |
| Cardiotoco. | 60.36/72.49 | 57.32/64.49 | 66.19/75.22 | 46.02/51.49 | 44.40/61.01 | 59.55/64.78 | 69.93/80.01 | **73.34/81.17** |
| Fraud | **69.39/96.30** | 40.46/95.73 | 34.90/95.48 | 33.20/95.19 | 0 .99/72.71 | 59.60/91.07 | 51.41/93.57 | 65.48/94.25 |
| Glass | 9 .52/57.71 | 9 .23/56.20 | 8 .96/54.80 | 9 .12/55.66 | 10.19/59.82 | 25.73/83.50 | 19.05/72.25 | **50.67/86.32** |
| Ionosphere | 97.68/96.83 | 95.91/94.54 | 89.69/87.65 | 86.70/85.52 | 70.46/70.41 | 97.77/97.10 | **98.02**/97.26 | 97.96/**97.34** |
| Mammo. | 33.34/82.20 | 40.63/89.22 | 41.78/90.03 | 41.90/88.79 | 11.41/74.12 | 18.94/65.48 | 47.55/**90.53** | **54.32**/89.87 |
| NSL-KDD | 75.32/73.88 | 74.50/54.96 | 75.29/57.07 | 79.35/75.51 | 75.04/61.36 | 51.94/16.84 | **90.85/87.80** | 87.20/85.18 |
| Optdigits | 15.70/82.39 | 43.63/96.65 | 6 .92/63.38 | 11.59/76.03 | 5 .36/47.06 | 16.96/7 .87 | **88.85/99.47** | 84.32/97.61 |
| Pima | 66.62/67.37 | 69.70/69.13 | 70.08/71.33 | 71.65/73.48 | 59.56/61.08 | 69.65/67.27 | **73.89/76.39** | 70.51/74.36 |
| Pendigits | 51.33/96.66 | 78.55/99.05 | 51.78/96.36 | 6 .16/45.63 | 4 .41/39.82 | 40.39/91.42 | 82.58/99.19 | **86.43/99.24** |
| Satellite | 85.83/80.26 | 80.88/73.91 | 77.78/66.63 | 82.17/76.59 | 68.66/72.59 | **87.99/85.49** | 85.32/79.62 | 84.95/82.00 |
| Satimage-2 | 88.46/99.38 | 96.92/99.61 | 91.92/98.17 | 94.27/98.81 | 11.42/89.94 | 81.24/97.92 | **98.50/99.92** | 97.54/99.83 |
| Shuttle | 91.72/99.61 | 96.01/99.83 | 94.88/99.69 | 98.18/99.52 | 48.75/90.49 | 98.11/99.35 | 98.79/99.75 | **98.63/99.92** |
| Thyroid | 60.55/92.71 | 78.92/98.56 | 81.34/98.55 | 72.82/98.87 | 10.95/71.49 | 65.75/95.18 | 84.17/98.04 | **88.37/99.21** |
| Wbc | 85.73/97.15 | 84.12/96.70 | 83.91/96.67 | 83.40/96.33 | 29.59/79.99 | 72.18/90.80 | 88.87/98.14 | **94.72/99.24** |
| Wine | 24.58/65.71 | 12.53/40.83 | 14.24/4 .85 | 14.76/50.67 | 49.07/88.33 | 56.59/91.50 | 93.35/95.38 | **99.99/100.0** |
| Average | 59.57/83.01 | 63.15/82.15 | 59.94/81.01 | 55.51/77.78 | 34.45/68.08 | 60.27/81.55 | 74.86/90.44 | **78.64/91.19** |

Table 2: Anomaly classification results (AUROC) on CIFAR-10 datasets and anomaly classification (image-level AUROC) & localization (pixel-level AUROC) results on MVTec AD dataset. The best and second results are highlighted in **bold/underline**.

| Normal Class | Airplane | Auto Mobile | Bird | Cat | Deer | Dog | Frog | Horse | Ship | Truck | ID Method | MVTec AD | |
|---|---|---|---|---|---|---|---|---|---|---|---|---|---|
| DSVDD[38] | 61.7 | 65.9 | 50.8 | 59.1 | 60.9 | 65.7 | 67.7 | 67.3 | 75.9 | 73.1 | PSVDD[52] | 92.1 | 95.7 |
| OCGAN[32] | 75.7 | 53.1 | 64.0 | 62.0 | 72.3 | 62.0 | 72.3 | 57.5 | 82.0 | 55.4 | PaDiM[11] | 95.8 | 97.5 |
| DROCC[15] | 82.1 | 64.8 | 69.2 | 64.4 | 72.8 | 66.5 | 68.6 | 67.5 | 79.3 | 60.6 | CutPaste[28] | 96.1 | 96.0 |
| HRN-L2[24] | 80.6 | 48.2 | 64.9 | 57.4 | _73.3_ | 61.0 | 74.1 | 55.5 | 79.9 | 71.6 | DRAEM[56] | 98.0 | 97.3 |
| HRN[24] | 77.3 | 69.9 | 60.6 | 64.4 | 71.5 | 67.4 | 77.4 | **82.5** | 77.3 | RD4AD[12] | 98.5 | 97.8 |
| PLAD[7] | _82.5_ | _80.8_ | 68.8 | 65.2 | 71.6 | _71.2_ | 76.4 | 73.5 | 80.6 | _80.5_ | PatchCore[37] | _99.1_ | 98.1 |
| DO2HSC[57] | 81.3 | **82.7** | **71.3** | _71.2_ | 72.9 | **72.8** | _83.0_ | _75.5_ | **84.4** | **82.0** | CFlow[17] | 98.3 | **98.6** |
| Ours | **83.1** | 78.4 | _69.8_ | **73.4** | **75.1** | 70.9 | **85.2** | **76.2** | 82.3 | 80.2 | Ours | **99.5** | _98.3_ |

from a prior distribution, achieving superior performance. Compared to the reconstruction-based work [53], our method improves AP by **+3.78** and AUROC by **+0.75**, benefiting from the proposed forward reconstruction that transforms abnormal inputs toward the normal manifold.

**Visual OCC Task**  Tab.2 shows the performance of our model on the natural image dataset, CIFAR-10, and the industrial defect detection dataset, MVTec AD. For CIFAR-10, we follow the standard OCC setting, where only one class is selected as the normal one, and the remaining classes are treated as abnormal. Under this setting, our method achieves the best performance compared to previous SOTA methods [38, 32, 15, 24, 7, 57], particularly when '**airplane**', '**cat**','**deer**', '**frog**', and '**horse**' are used as the normal class. For industrial data, our method also achieves competitive performance compared to widely used methods [52, 11, 28, 56, 12, 37, 17]. Specifically, compared to the reconstruction-based method [12], our method improves image-level and pixel-level AUROC by **+1.0** and **+0.5**, respectively, as the proposed contraction mapping can constrain reconstructed results to the normal manifold to enhance the performance.

## 4.2 Ablation Study

In this section, we investigate the contributions of the main components and impact of key hyperparameters of our approach on both tabular and visual datasets.

Table 3: Ablation study of our method on tabular datasets using AP/AUROC metrics. We show results on 5 selective datasets and mean results on 20 used datasets. 'RE': reconstruction via contraction mapping. 'DE': density estimation via reversed flow. The best result is highlighted in **bold**.

| Model ↓ | RE | DE | Cardio | Glass | Thyroid | Wbc | Mammo. | All Tabular Datasets |
|---------|----|----|--------|-------|---------|-----|--------|----------------------|
| **A** | – | – | 80.84/93.71 | 15.30/71.08 | 82.31/98.17 | 71.21/93.53 | 31.42/87.21 | 67.44/86.52 |
| **B** | ✓ | – | 84.25/95.15 | 36.77/85.65 | 86.01/99.08 | 88.98/97.18 | 54.02/89.38 | 75.21/90.32 |
| **C** | – | ✓ | 76.64/92.11 | 39.63/81.87 | 82.83/98.65 | 90.04/98.35 | 52.76/86.64 | 73.59/88.05 |
| **D** | ✓ | ✓ | **86.35/96.21** | **50.67/86.32** | **88.37/99.21** | **94.72/99.24** | **54.32/89.87** | **78.64/91.19** |

**Effectiveness of Main Components**  Tab.3 demonstrates the performance of four model variants: **A.** the baseline model with a six-layer MLP (equivalent to the number of flow-based iterations for fair comparison) for the reconstruction; **B.** A reconstruction-only variant using the proposed contraction mapping flow; **C.** A density-estimation-only variant using the flow structure; **D.** Our full method combines forward reconstruction and backward density estimation. The model **B** significantly outperforms the baseline method, which validates the effectiveness of the reconstruction via the contraction mapping. In addition, model **C** also surpasses the baseline, demonstrating the effectiveness of density estimation via the flow structure. When combining both, our method achieves AP/AUROC by **78.64/91.9**, obtaining a gain of **+11.2/4.67** compared to our baseline. Tab.4 shows an ablation study on visual data, where our framework improves our baseline by **+4.87** on CIFAR-10 and **+1.7/1.1**. To this end, the effectiveness of our method is demonstrated in the general OCC-based AD task (visual AD and tabular AD tasks).

Table 4: Ablation study of our method on CIFAR-10 and MVTec AD, where image-level AUROC is used for the former and image/pixel-level AUROC for the latter. The best result is highlighted in **bold**.

| Model ↓ | RE | DE | CIFAR-10 | MVTec AD |
|---------|----|----|----------|----------|
| **A** | – | – | 72.59 | 97.8/97.2 |
| **B** | ✓ | – | 76.88 | 99.2/97.9 |
| **C** | – | ✓ | 74.27 | 98.1/97.5 |
| **D** | ✓ | ✓ | **77.46** | **99.5/98.3** |

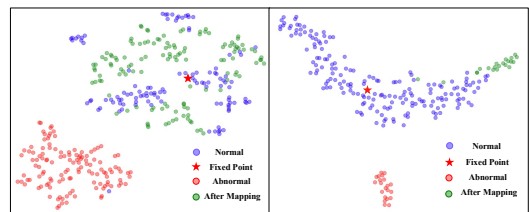

Figure 4: T-SNE visualization of feature distribution after CM with $R = 3$ and $K = 2$ on 'Ionosphere' (left) and 'Wbc' (right) datasets.

Table 5: Impact of hyperparameters $K$ & $R$ in our method, where all tabular and MVTec AD datasets are adopted for ablation. AP/AUROC are leveraged for tabular data and image-level/pixel-level AUROC for industrial data. The best result is highlighted in **bold**.

| $\frac{K \longrightarrow}{R \downarrow}$ | $K = 1$ | $K = 2$ | $K = 3$ | $K = 4$ | **Visual AD** **Base: 97.8/97.2** | $K = 1$ | $K = 2$ | $K = 3$ | $K = 4$ |
|---|---|---|---|---|---|---|---|---|---|
| $R = 1$ | 60.42/78.75 | 64.81/83.20 | 68.87/85.20 | 67.04/85.33 | $R = 1$ | 97.4/96.8 | 97.7/97.2 | 98.3/97.6 | 98.2/97.7 |
| $R = 2$ | 64.34/82.11 | 74.89/88.21 | 75.21/88.95 | 73.44/87.59 | $R = 2$ | 98.3/97.4 | 98.9/98.0 | 99.1/97.9 | 99.2/97.9 |
| $R = 3$ | 71.21/86.32 | **78.64/91.19** | 76.42/89.94 | 70.87/85.12 | $R = 3$ | 98.4/97.4 | 99.3/98.3 | **99.5/98.3** | 99.1/98.1 |
| $R = 4$ | 74.57/89.36 | 77.03/90.55 | 73.14/87.60 | 73.25/86.47 | $R = 4$ | 98.7/97.8 | 99.2/98.1 | 98.9/97.9 | 98.9/97.7 |
| $R = 5$ | 75.38/90.29 | 74.08/88.76 | 68.27/84.88 | 63.44/81.09 | $R = 5$ | 98.6/97.7 | 99.0/98.0 | 98.5/97.5 | 98.3/97.3 |

**Impact of Hyperparameters in Contraction Mapping Flow**  Tab.5 shows the impact of hyperparameters on 20 tabular datasets and MVTec AD. In the tabular AD task, when $R = 1$ and $K = 1$, due to the limited depth of $\varphi(\cdot)$ results in fewer parameters and insufficient contraction mapping iterations, leading to suboptimal performance. When $K = 2$ and $R = 3$, our framework achieves the best performance, with an AP/AUROC of **78.64/91.19**. However, when $K \times R$ becomes too large, excessive iterations can cause reconstructions to converge too closely to the fixed point, degrading the performance. For the industrial image-based AD task, when $K = 3$ and $R = 3$, the best performance of **99.5/98.3** in terms of image/pixel-level AUROC is achieved. Both large and small values of $K \times R$ may degrade the anomaly classification and localization. We provide a T-SNE visualization of the feature distribution after CM with $R = 3$ and $K = 2$ in Fig.4, where abnormal representations are constrained within the normal manifold and are prevented from fully converging to the fixed point.

## 5 Conclusion

Recent reconstruction-based methods trained only on normal samples may generalize to certain abnormal inputs, leading to well-reconstructed anomalies and degraded detection performance. To address this issue, we propose to apply the contraction mapping structure to conduct reconstruction, in which normal and abnormal inputs are transformed into the normal manifold via iterations of this mapping to increase the reconstruction error. In addition, the contraction mapping is also designed to be invertible, which means it can estimate the likelihood of the original input based on a Gaussian model in the flow form. Based on the above two points, we propose a bidirectional structure with forward reconstruction and backward density estimation. Extensive experiment evaluations on tabular data, natural image data, and industrial image data demonstrate the effectiveness of our method.

## 6 Acknowledgments

This work is supported by the National Natural Science Foundation of China (No. 62471405, 62331003, 62301451), Jiangsu Basic Research Program Natural Science Foundation (BK20241814), Natural Science Foundation of Hebei Province (F2024105029), Suzhou Basic Research Program (SYG202316) and XJTLU REF-22-01-010, XJTLU AI University Research Center, Jiangsu Province Engineering Research Center of Data Science and Cognitive Computation at XJTLU and SIP AI innovation platform (YZCXPT2022103).

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

# A More Implementation Details

In this section, we provide additional implementation details, including the hyperparameters and network architecture for feature extraction.

**Hyperparameters for Tabular Datasets**  Tab.6 of the Appendix shows details of 20 tabular anomaly detection (AD) datasets used in the experiment, where different datasets contain significantly different numbers of samples and feature dimensions. It is challenging to set global training hyperparameters to achieve optimal performance across all datasets. To ensure good performance while avoiding excessive hyperparameter tuning, we fix the learning rate as well as the dimension of input and output of the CM across all datasets (see Tab.7). Finally, following previous works [53], we only adjust the batch size and epochs during training for datasets with particularly large or small sample sizes to achieve better performance.

Table 6: Detailed information of 20 tabular datasets. **N**: the total number of samples, **D**: the number of attributes in each dataset.

| Dataset | N | D | Anomalies |
|---|---|---|---|
| Arrhythmia | 452 | 274 | 66 (14.6%) |
| Breastw | 683 | 9 | 239 (34.9%) |
| Cardio | 1,831 | 21 | 176 (9.6%) |
| Census | 299,285 | 500 | 18,568 (6.2%) |
| Campaign | 41,188 | 62 | 4,640 (11.2%) |
| Cardiotocography | 2,114 | 21 | 466 (22.0%) |
| Fraud | 284,807 | 29 | 492 (0.1%) |
| Glass | 214 | 9 | 9 (4.2%) |
| Ionosphere | 351 | 33 | 126 (35.8%) |
| Mammography | 11,183 | 6 | 260 (2.3%) |
| NSL-KDD | 148,517 | 122 | 77,054 (51.8%) |
| Optdigits | 5,216 | 64 | 150 (2.8%) |
| Pima | 768 | 8 | 268 (34.8%) |
| Pendigits | 6,870 | 16 | 156 (2.2%) |
| Satellite | 6,435 | 36 | 2,036 (31.6%) |
| Satimage-2 | 5,803 | 36 | 71 (1.2%) |
| Shuttle | 49,097 | 9 | 3,511 (7.1%) |
| Thyroid | 3,772 | 6 | 93 (2.4%) |
| Wbc | 278 | 30 | 21 (7.5%) |
| Wine | 129 | 13 | 10 (7.7%) |

Table 7: Traning hyperparameters for 20 tabular data. '**Lr**': learning rate. '**BS**': batch size. '**C**': the dimension of input and output of CM.

| Dataset | Lr | Epoch | BS | C |
|---|---|---|---|---|
| Arrhythmia | 1e-5 | 100 | 64 | 32 |
| Breastw | 1e-5 | 100 | 64 | 32 |
| Cardio | 1e-5 | 100 | 64 | 32 |
| Census | 1e-5 | 150 | 512 | 32 |
| Campaign | 1e-5 | 150 | 512 | 32 |
| Cardiotocography | 1e-5 | 150 | 64 | 32 |
| Fraud | 1e-5 | 150 | 512 | 32 |
| Glass | 1e-5 | 100 | 64 | 32 |
| Ionosphere | 1e-5 | 150 | 64 | 32 |
| Mammography | 1e-5 | 100 | 512 | 32 |
| NSL-KDD | 1e-5 | 100 | 512 | 32 |
| Optdigits | 1e-5 | 150 | 64 | 32 |
| Pima | 1e-5 | 150 | 64 | 32 |
| Pendigits | 1e-5 | 150 | 64 | 32 |
| Satellite | 1e-5 | 150 | 256 | 32 |
| Satimage-2 | 1e-5 | 150 | 256 | 32 |
| Shuttle | 1e-5 | 150 | 256 | 32 |
| Thyroid | 1e-5 | 150 | 256 | 32 |
| Wbc | 1e-5 | 150 | 64 | 32 |
| Wine | 1e-5 | 150 | 64 | 32 |

**Feature Extractor**  Following previous work [57, 53], we also adopt a three-layer MLP with LeakyReLU activation to extract the deep representation of tabular data (see Fig.5 (a)), where the dimension of the output and input in each layer is consistent with previous work [57]. Next, as shown in Fig.5 (b), we leverage a simple three-layer CNN to extract the deep representation of natural image data, where the setting of the network follows work [57].

**Network Details for Industrial Defect Detection**  Industrial defect detection is a primary application area of anomaly detection that has a well-established standard. Therefore, we adopt a baseline model to obtain a baseline performance (see Tab.5 in our main manuscript). The baseline model adopts a frozen encoder and a learnable decoder, which is similar to RD4AD [12]. Unlike RD4AD, we removed the MMF and OCE modules from our baseline framework, resulting in a more fundamental performance. Based on this baseline framework, the contraction mapping flow $\varphi(\cdot)$ is applied after the feature encoder (see Fig.5 (c)). The specific pipeline can be summarized as: given an image $\mathbf{I} \in \mathbb{R}^{H_0 \times W_0 \times 3}$, the frozen encoder $\mathcal{E}(\cdot)$ encodes $\mathbf{I}$ into multi-layer features $\{\mathbf{x}_l\}_{l=1}^{L}$, each $\mathbf{x}_l \in \mathbb{R}^{H_l \times W_l \times C_l}$ denotes the $l$-th layer feature. We select $l_1$-th, $l_2$-th, $l_3$-th layer features as encoded features, denoted by $\{\mathbf{x}_1, \mathbf{x}_2, \mathbf{x}_3\}$. Next, we down-sample the size of $\mathbf{x}_1, \mathbf{x}_2$ to $H_{l_3} \times W_{l_3}$. Next, we concatenate encoded features in the channel dimension, denoted by $[\mathbf{x}_1, \mathbf{x}_2, \mathbf{x}_3] \in \mathbb{R}^{H_{l_3} \times W_{l_3} \times (C_{l_1} + C_{l_2} + C_{l_3})}$, and apply a linear projection $\Phi(\cdot)$ to fuse them, i.e., $\mathbf{x} = \Phi([\mathbf{x}_1, \mathbf{x}_2, \mathbf{x}_3]) \in \mathbb{R}^{H_{l_3} \times W_{l_3} \times C_{l_3}}$. Then, $\mathbf{x}$ is input to our contraction mapping flow $\varphi(\cdot)$, obtaining latent variable $\mathbf{z}$. Under the reconstruction branch, we apply a learnable decoder to decode $\mathbf{z}$ into decoded features $\{\widehat{\mathbf{x}}_1, \widehat{\mathbf{x}}_2, \widehat{\mathbf{x}}_3\}$. We apply the cosine similarity loss to replace the original $\mathcal{L}_{rec}$

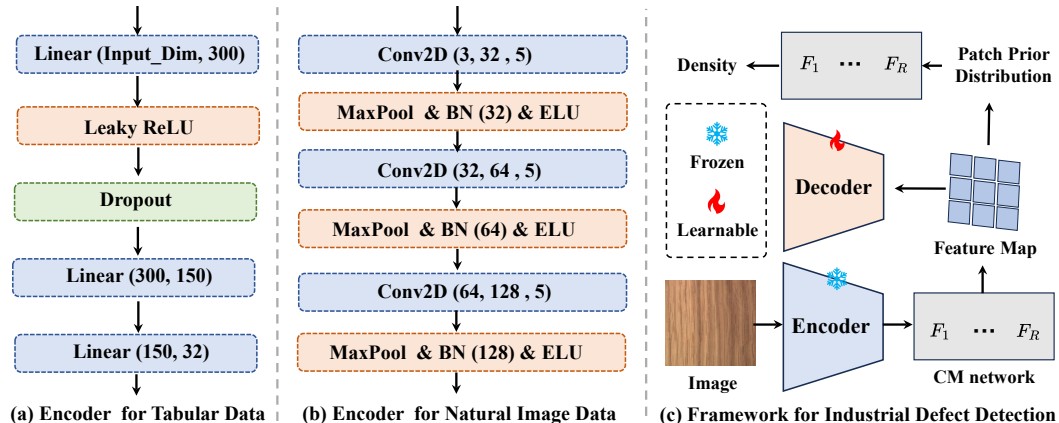

**(a) Encoder for Tabular Data**    **(b) Encoder for Natural Image Data**    **(c) Framework for Industrial Defect Detection**

Figure 5: Different feature extractors are used to extract deep representations of different data types. (a): Three-layer MLP with LeakyReLU activation is adopted for tabular data. (b): Three-layer CNN with Batch normalization and ELU activation is used for natural image data. (c): Our proposed method is based on a general baseline method for industrial defect detection.

(Eq.(1) in the main manuscript), i.e.,

$$\mathcal{L}_{\text{rec}} = \sum_{i=1}^{3} \left( 1 - \frac{\mathbf{Flat}(\mathbf{x}_i) \cdot \mathbf{Flat}(\widehat{\mathbf{x}}_i)}{\|\mathbf{Flat}(\mathbf{x}_i)\|_2 \|\mathbf{Flat}(\widehat{\mathbf{x}}_i)\|_2} \right), \tag{9}$$

where $\mathbf{Flat}(\cdot)$ is the flatten operator. The density estimation branch adopts patch embedding features to fit the Gaussian prior distribution $\pi(\cdot)$ and uses $\varphi^{-1}(\cdot)$ to obtain the likelihood of each patch embedding feature. During inference, we adopt a combination of both the cosine distance and the likelihood of each patch embedding feature as the patch-level anomaly score. Through the optimization of Eq. (9), contraction mapping flow $\varphi(\cdot)$ can transform any input $\mathbf{x}$ onto the normal manifold, which ensures that the decoded features $\{\widehat{\mathbf{x}}_1, \widehat{\mathbf{x}}_2, \widehat{\mathbf{x}}_3\}$ are on the normal manifold, increasing the reconstruction error.

## B  More Contraction Mapping Network Details

According to **Section** 3.1.2 in the main manuscript, the structure of contraction mapping can be written as:

$$\varphi(\cdot) = F_R \circ F_{R-1} \cdots F_1(\cdot),$$

where each $F_r(\cdot) = f_K^r \circ f_{K-1}^r \circ \cdots f_1^r(\cdot)$. Each contraction mapping unit $f_k^r(\cdot)$ is based on the 1-Lipchitz invertible neural network $\mathbf{G}$, i.e.,

$$f(x) = \left( \frac{\mathbf{Id} + \mathbf{G}}{2} \right)^{-1} (x) - x,$$

where $\mathbf{Id}$ denotes the identity function. We adopt i-DenseNet [33] to parametrize $\mathbf{G}$. Following [1, 2, 31], $\mathbf{G}$ is designed as a Lipschitz-continuous function with Lipschitz constant $\eta < 1$, where $\|\mathbf{G}(x_1) - \mathbf{G}(x_2)\|_2 \leq \eta \|x_1 - x_2\|_2$. Based on $\mathbf{G}$, $f(\cdot)$ is monotonic with Lipschitz constant $\eta_f < 1$, so $f(\cdot)$ is an invertible mapping according to Banach fixed point theorem [16] (the proof can be found in [1]). Based on $\mathbf{G}$ and Eq. (2), we can model $f_k^r(\cdot)$, which is named as 'Lipschitz Layer'. We then apply the LipSwish activation function to the output of each $f_k^r(\cdot)$. The LipSwish function is written as:

$$\text{LipSwish}(x) = \frac{x}{1.1 * (1 + \exp(-\beta x))},$$

## C  Supplementary Proof

In this section, we supplement the detailed proofs of Lemma 1 and Lemma 2 used in our main manuscript.

**Lemma 3** *(Convergence of Iterations): Let $f(\cdot)$ be a contraction mapping with $\eta \in [0,1)$. For any initial input $x_0 \in \mathbb{R}^d$, we define the sequence $\{x_n\}$ by $x_{n+1} = f(x_n)$ for $n = 1,2,\cdots$. Then $\lim_{n\to\infty} x_n = x^*$, where $x^*$ is unique and is called a fixed point of $f(\cdot)$.*

**proof.** *We firstly prove that $\{\mathbf{x}_n\}$ is a Cauchy sequence. For any $m > n$, we have:*

$$\|x_m - x_n\|_2 = \|x_n - x_m\|_2$$
$$\leq \|x_n - x_{n+1}\|_2 + \|x_{n+1} - x_{n+2}\|_2 + \ldots + \|x_{m-1} - x_m\|_2.$$

*Next, for each term $\|x_{n+k} - x_{n+k+1}\|_2$ in this sum:*

$$\|x_{n+k} - x_{n+k+1}\|_2 = \|f(x_{n+k-1}) - f(x_{n+k})\|_2$$
$$\leq \eta \cdot \|x_{n+k-1} - x_{n+k}\|_2$$
$$\leq \eta^2 \cdot \|x_{n+k-2} - x_{n+k-1}\|_2$$
$$\vdots$$
$$\leq \eta^{n+k} \cdot \|x_0 - x_1\|_2.$$

*Therefore, based on the above two inequality, we have:*

$$\|x_m - x_n\|_2 \leq \sum_{k=0}^{m-n-1} \|x_{n+k} - x_{n+k+1}\|_2 \leq \sum_{k=0}^{m-n-1} \eta^{n+k} \cdot \|x_0 - x_1\|_2$$

$$= \eta^n \cdot \|x_0 - x_1\|_2 \cdot \sum_{k=0}^{m-n-1} \eta^k = \eta^n \cdot \|x_0 - x_1\|_2 \cdot \frac{1 - \eta^{m-n}}{1 - \eta}$$

$$\leq \eta^n \cdot \|x_0 - x_1\|_2 \cdot \frac{1}{1 - \eta}.$$

*As $n \to \infty$, $\eta^n \to 0$ (since $\eta < 1$), so $\|x_m - x_n\| \to 0$ for any $m > n$ as $n \to \infty$. This proves that $\{x_n\}$ is a Cauchy sequence. Since $\mathbb{R}^d$ with the Euclidean metric is complete, the sequence converges to a certain point $x^* \in \mathbb{R}^d$, i.e. $\lim_{n\to\infty} x_n = x^*$. Next, by continuity of $f$ (which follows from the contraction property) and the definition of $\{x_n\}$:*

$$f(x^*) = f(\lim_{n\to\infty} x_n) = \lim_{n\to\infty} f(x_n) = \lim_{n\to\infty} x_{n+1} = x^*.$$

*To this end, we prove that $f(x^*) = x^*$, i.e., $x^*$ is a unique fixed point of $f(\cdot)$.*

**Lemma 4** *Let $\varphi_K(\cdot) = f_K \circ f_{K-1} \circ \cdots f_1(\cdot)$, where each $f_k$ is contraction mapping with $\eta_k \in [0,1)$, then $\varphi_K(\cdot)$ is a contraction mapping with Lipschitz constant $\eta_\varphi \leq \eta_1 \cdot \eta_2 \cdots \eta_K$.*

**proof.** *First of all, let us prove for a base case with $K = 2$. Let $\varphi_2(\cdot) = f_2 \circ f_1(\cdot)$ be the composition of two contraction mappings. For any two points $x_1, x_2 \in \mathbb{R}^d$, we have:*

$$\|\varphi_2(x_1) - \varphi_2(x_2)\|_2 = \|f_2(f_1(x_1)) - f_2(f_1(x_2))\|_2.$$

*Since $f_2$ is a contraction mapping with constant $\eta_2$:*

$$\|f_2(f_1(x_1)) - f_2(f_1(x_2))\|_2 \leq \eta_2 \|f_1(x_1) - f_1(x_2)\|_2. \tag{10}$$

*Since $f_1$ is a contraction mapping with constant $\eta_1$:*

$$\eta_2 \|f_1(x_1) - f_1(x_2)\|_2 \leq \eta_2 \cdot \eta_1 \|x_1 - x_2\|_2. \tag{11}$$

*Base on the above Eq. (10) and Eq. (11), we have:*

$$\|\varphi_2(x_1) - \varphi_2(x_2)\|_2 \leq \eta_2 \cdot \eta_1 \|x_1 - x_2\|_2,$$

*which proves that $\varphi_2(\cdot)$ is contraction mapping with $\eta_\varphi \leq \eta_1 \cdot \eta_2$. Next, we apply mathematical induction to prove the general case. We assume the lemma holds for $K-1$ mappings, i.e., $\varphi_{K-1}(\cdot) = f_{K-1} \circ \cdots \circ f_1(\cdot)$ is a contraction mapping with Lipschitz constant $\eta_{\varphi_{K-1}} \leq \eta_1 \cdot \eta_2 \cdots \eta_{K-1}$. We consider $\varphi_K(\cdot) = f_K \circ \varphi_{K-1}(\cdot)$. For any $x_1, x_2 \in \mathbb{R}^d$, we have:*

$$\|\varphi_K(x_1) - \varphi_K(x_2)\|_2 = \|f_K(\varphi_{K-1}(x_1)) - f_K(\varphi_{K-1}(x_2))\|_2. \tag{12}$$

*Then, based on the inductive hypothesis and the definition of contraction mapping, we have:*

$$\eta_K \|\varphi_{K-1}(x_1) - \varphi_{K-1}(x_2)\|_2 \leq \eta_K (\eta_1 \cdot \eta_2 \cdots \eta_{K-1}) \|x_1 - x_2\|_2. \tag{13}$$

*Combining Eq. (12) and Eq. (13), one inequality can be obtained:*

$$\|\varphi_K(x_1) - \varphi_K(x_2)\|_2 \leq (\eta_1 \cdot \eta_2 \cdots \eta_K) \|x_1 - x_2\|_2.$$

*Since each $\eta_k \in [0,1)$, $\eta_\varphi$ must also be in $[0,1)$, i.e., $0 \leq \eta_1 \cdot \eta_2 \cdots \eta_K < 1$.*

## D   Supplementary Experiments

**Sensitivity of Balancing the Loss Terms**   The hyperparameter $\lambda$ in Eq.(7) of the main manuscript balances the reconstruction loss $\mathcal{L}_{\mathrm{rec}}$ and the maximum likelihood estimation loss $\mathcal{L}_{\mathrm{MLE}}$. To investigate its impact, we vary $\lambda$ and evaluate the performance, as shown in Tab. 8 of the appendix. For the 20 tabular datasets, $\lambda = 0.5$ yields the best AP/AUROC, **78.64/91.19**. Slightly higher ($\lambda = 0.6$) or lower ($\lambda = 0.4$) values show comparable but slightly degraded performance, indicating that the performance is stable around $\lambda = 0.5$. For the MVTec AD dataset, $\lambda = 0.4$ provides the highest image-level AUROC, while $\lambda = 0.5$ achieves the best pixel-level AUROC. Too small $\lambda$ causes $\varphi^{-1}(\cdot)$ to not fully capture the real distribution of input, degrading the performance of density estimation. Too large a value of $\lambda$ may disturb reconstruction training. Therefore, the choice of $\lambda = 0.5$ (as used for both tabular and MVTec AD in our main experiments) achieves robust performance across both types of datasets.

Table 8: Anomaly detection results when varying different $\lambda$ values for balancing $\mathcal{L}_{\mathrm{rec}}$ and $\mathcal{L}_{\mathrm{MLE}}$. We adopt image-level AUROC metric for 20 tabular datasets and image/pixel-level AUROC for industrial MVTec AD datasets. **Bold**/underline values indicate the best/runner-up.

| Datasets | $\lambda = 0.2$ | $\lambda = 0.3$ | $\lambda = 0.4$ | $\lambda = 0.5$ | $\lambda = 0.6$ | $\lambda = 0.7$ | $\lambda = 0.8$ |
|---|---|---|---|---|---|---|---|
| Tabular Data | 77.86/91.12 | 78.29/91.16 | 78.19/91.14 | **78.64**/91.19 | 78.55/**91.21** | 78.35/91.09 | 78.13/91.10 |
| MVTec AD | 99.1/98.1 | 99.3/98.1 | **99.6**/98.2 | 99.5/**98.3** | 99.3/98.2 | 99.4/98.3 | 98.9/97.9 |

**Sensitivity of Weight Coefficient $\alpha$**   The weight coefficient $\alpha$ in the anomaly score (Eq.(8) in the main manuscript) controls the contribution of the log-likelihood term relative to the reconstruction error. We evaluated the effect of different $\alpha$ values, with results presented in Tab. 9 of the Appendix. We note that the scale of reconstruction error is generally a very small value, while the scale of log likelihood is much larger than the reconstruction error. Therefore, using $\alpha$ to control the scale log likelihood is necessary for our unified framework that combines these two different paradigms. For the 20 tabular datasets, an $\alpha$ of 0.01 achieves the optimal performance by 78.64/91.19. Performance degrades if $\alpha$ is too large or too small because of disturbed reconstruction or density estimation. Similarly, for the MVTec AD dataset, $\alpha = 0.001$ also results in the best image-level and pixel-level AUROC. Again, performance drops with much larger or smaller $\alpha$ values. This analysis highlights that the density estimation component, when weighted appropriately by $\alpha$, significantly contributes to the final anomaly score. An $\alpha$ value of 0.001 appears to be a generally effective choice for both types of datasets according to this ablation.

Table 9: Anomaly detection results with different weight coefficient $\alpha$ in anomaly score. We adopt the image-level AUROC metric for 20 tabular datasets and the image/pixel-level AUROC for industrial MVTec AD datasets. **Bold**/underline values indicate the best/runner-up.

| $\alpha$ | $\alpha = 1$ | $\alpha = 0.5$ | $\alpha = 0.1$ | $\alpha = 0.05$ | $\alpha = 0.01$ | $\alpha = 0.005$ | $\alpha = 0.001$ |
|---|---|---|---|---|---|---|---|
| Tabular Data | 74.48/88.62 | 76.40/89.71 | 77.26/90.97 | 78.05/91.09 | **78.64/91.19** | 78.21/91.15 | 76.89/90.88 |
| $\alpha$ | $\alpha = 0.1$ | $\alpha = 0.05$ | $\alpha = 0.01$ | $\alpha = 0.005$ | $\alpha = 0.001$ | $\alpha = 0.0005$ | $\alpha = 0.0001$ |
| MVTec AD | 98.3/97.6 | 98.6/97.6 | 98.9/98.0 | 99.3/98.2 | **99.5/98.3** | 99.3/98.1 | 99.2/98.0 |

## E   Limitations

While our method shows strong AD performance on tabular and image datasets, its current scope is primarily focused on these common data types. It has not yet been extended to critical domains like graph and time-series data, where optimal AD typically requires specialized network architectures to handle their unique structural and dependency characteristics. Future work will involve adapting our URD framework to these distinct modalities by incorporating tailored design principles, aiming to progress toward a unified anomaly detection model.

# F   Some Questions and Answers regarding Motivation and Methodology

## F.1   Intuitive Understanding of Proposition 1

Proposition 1 mainly focuses on establishing a direct causal link between our optimization and the fixed point's location. The proposition states that $\|\mathbf{x}^* - \mathbf{x}\|_2 \leq \mathcal{L}_{\text{rec}}/(1-\eta)$, which directly and rigorously demonstrates that by minimizing the reconstruction loss $\mathcal{L}_{\text{rec}}$ on normal samples x, we are guaranteed to bound the distance to the fixed point $\mathbf{x}^*$.

## F.2   Why Apply LipSwish Activation Function rather than ReLU?

We note that the choice of LipSwish is deliberate and critical. LipSwish guarantees the reversibility of the entire network, while ReLU cannot guarantee it. In this work, we apply an invertible network $\varphi$ as the foundation, where LipSwish activation function is invertible and ensures the invertibility of $\varphi$. In contrast, $\mathbf{ReLU}(x) = \mathbf{Max}(0, x)$ maps all negative inputs to zero, which creates a many-to-one mapping that is mathematically non-invertible. We adopt i-DenseNet [33] to parameterize our mapping $\mathbf{G}$ in Eq.(2). According to [33], using LipSwish is a standard and effective choice in this context, providing more expressive power than simpler activations. LipSwish is also smooth, continuously differentiable, and invertible, which are essential properties for stable training of INNs.

## F.3   How does Our Contraction Mapping Work in Different Data Distributions?

Since the contraction mapping iteratively transforms data toward a fixed point derived by training the model on normal data, what if the normal data distribution itself is chaotic? Or if the normal data distribution is ring-shaped or U-shaped? Can the fixed point be located in an appropriate position? Will the reconstruction discrepancy identification work well in such cases?

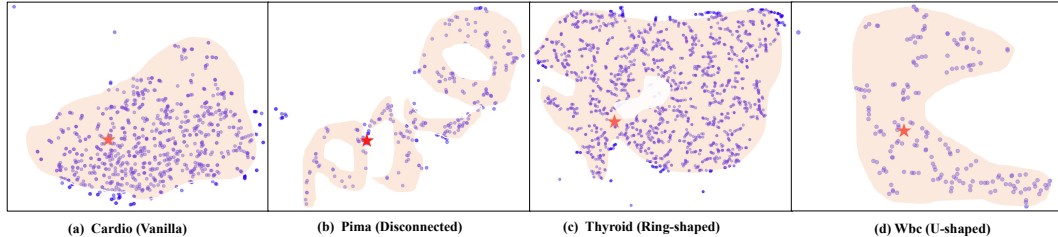

(a) Cardio (Vanilla)   (b) Pima (Disconnected)   (c) Thyroid (Ring-shaped)   (d) Wbc (U-shaped)

Figure 6: T-SNE visualization of data distribution shapes and corresponding fixed point positions for four types of tabular datasets

Sometimes, extreme and complex distributions may make the reconstruction-only part ineffective, which is also the reason why we introduce a density estimation process to collaborate with the reconstruction process. Following the above questions, we discussed several situations and provided special cases of difficult reconstruction.

**Vanilla Distribution.** In fact, the manuscript visualizes the position of the fixed point in different normal distributions. Generally, the fixed point is not completely located at the geometric center of the distribution, and sometimes, due to different data or training settings, fixed points may even fall on the edges of the distribution (see Fig.6 (a)).

**Disconnected Distribution.** As shown in Fig.6 (b), the normal distribution is relatively scattered and chaotic, and the fixed point is located at the edge of one of the sub distributions. However, because the abnormal distribution is far from the normal one, reconstruction error is still an effective criterion, achieving a good performance under a reconstruction-only setting.

**Ring-shaped Distribution.** There is currently no typical ring-shaped normal distribution in the used datasets. Fig.6 (b) and (c) show multi-ring and irregular-ring structures, respectively. In fact, through multiple adjustments to the training hyperparameters and random seed, we found that fixed points may be located in some data-free areas (central regions of the ring) or within the distribution (internal regions of the ring). If the anomaly is located in the central area of the ring, this method may theoretically fail, but this situation has not been found in experiments yet.

**U-shaped Distribution.** As shown in Fig.6 (d), the normal distribution is U-shaped and the fixed point is located at the bottom of 'U'. We found that after adjusting the random seed or training hyperparameters, the position of the fixed point will change and may not necessarily be located at the bottom position. However, the same thing is true: as long as enough training epochs are set, the fixed point is located inside or at the edge of U. CM will achieve good and stable performance as long as the abnormal data is in a distant external area.

In the reconstruction-only experiments, we found that if the abnormal data is very close to the normal distribution in terms of spatial position, the reconstruction errors generated by CM for normal and abnormal data are very close, resulting in false negatives and degrading performance. Of course, this is also a challenge for all reconstruction-based methods. In addition, since fixed points may appear at any position on a normal manifold, they may be closer to some anomalies than other normal points, leading to false negatives. Finally, in the most extreme case, our framework will fail when a few anomalies are within the normal distribution. However, it should be noted that almost all existing OCC methods cannot address this type of problem. This combined approach makes our method robust even for complex data distributions.

## F.4 Trade-off between Convergence to a Fixed Point and Bijectivity via Network Design

Our method focuses on leveraging a finite number of mapping compositions to simulate iteration of a single contraction mapping, rather than directly using iteration of a single contraction mapping.

Specifically, according to Proposition 2, we apply $\varphi_K(\cdot) = f_K \circ f_{K-1} \circ \cdots f_1(\cdot)$ to simulate a finite number of iterations of $f$, rather than directly iterating $f$, where each $f_k$ has its own parameters $\theta_k$ and Lipschitz constant $\eta_k$. Lipschitz constant $\eta_\varphi \leq \eta_1 \cdot \eta_2 \cdots \eta_K < 1$ ensures contracting anomalies towards the normal distribution. We leverage $\mathcal{L}_{\text{rec}} = \|\varphi_K(\mathbf{x}) - \mathbf{x}\|_2$ to train the entire network $\varphi_K$, rather than firstly training a single $f$ and then iterating it a few times. From an optimization perspective, $\varphi_K$ optimized by $\mathcal{L}_{\text{rec}}$ tends to restore normal input to normal output instead of contracting all inputs (normal and abnormal) to one fixed point. From a mathematical perspective, a finite and small $K$ value gives $\varphi_K$ the ability to contract the input to the normal manifold. Still, it does not destroy the reversibility of $\varphi_K$. In summary, selecting $K$ as a finite value, the contraction properties of $\varphi_K$ and the training objective jointly ensure that it can map anomalies to normal manifolds. Still, it will not completely shrink to a very small region or even a point and destroy its reversibility.

We intentionally keep $K$ and $R$ small in actual experiments to prevent over-contraction. This allows us to benefit from the "pull" of the contraction towards the normal manifold (which increases reconstruction error for anomalies) while not destroying the reversibility of our INN. The trade-off is managed by the choice of $K$ and $R$, which we study empirically in Table 5. The results show that a moderate number of iterations yields the best performance, validating our handling of this trade-off.

## F.5 The Risk of CM Network Collapsing to the Identity Mapping

The degradation towards identity mapping is a critical concern for contraction mapping trained with reconstruction loss. We show two explicit mechanisms to prevent this: one is the structure of the contraction mapping unit, and the other is the regularity of our backward density estimation.

**(1)** According to Eq.(2), $f(x) = (\frac{\mathbf{Id}+\mathbf{G}}{2})^{-1}(x) - x$, where $\mathbf{G}$ is invertible and $\mathbf{Id}$ is identity mapping. If $f$ degrades into an identity mapping under reconstruction loss, we have $(\frac{\mathbf{Id}+\mathbf{G}}{2})^{-1}(x) - x = x$, which means $(\frac{\mathbf{Id}+\mathbf{G}}{2})^{-1}(x) = 2x$. Based on the definition of inverse mapping, we must have $\frac{\mathbf{Id}+\mathbf{G}}{2}(2x) = x$. This means $\mathbf{Id}(2x) + \mathbf{G}(2x) = 2x$. Then, we can get $\mathbf{G}(2x) = 0$ to make $f(x)$ an identity mapping. Paradoxically, $\mathbf{G}$ is invertible and a one-to-one mapping. Therefore, in terms of the structure of $f$, it is difficult to degrade into an identity mapping under the optimization of reconstruction loss.

**(2)** The backward density estimation loss $\mathcal{L}_{\text{MLE}}$ provides a powerful regularizer against identity collapse. In the backward process, $\mathcal{L}_{\text{MLE}}$ pushes $\varphi^{-1}$ to be a mapping that transforms a basic Gaussian distribution $\pi(\mathrm{z})$ to the (typically complex) input distribution $p(\mathrm{x})$. An identity mapping could only achieve this if $p(\mathrm{x})$ is also a Gaussian distribution, which is never the case. This forces $\varphi^{-1}$ to learn a non-trivial, meaningful transformation.

### F.6 The Conflict of Optimization Objectives of Reconstruction and Density Estimation

According to reconstruction loss (1) and MLE loss (6), pushes z to be close to x, while $\mathcal{L}_{\text{MLE}}$ pushes the distribution of z to be a Gaussian distribution. We need to clarify that both forward reconstruction (RE) and backward density estimation (DE) are indispensable, although they may seem contradictory. We have also designed corresponding trade-off schemes in the manuscript.

When only using RE, if some anomalies are very close to the normal manifold or the obtained fixed point in terms of spatial position, this causes small reconstruction errors and false negatives. In addition, the distribution of normal data may be very complex, chaotic, and disconnected (see Fig.6). In this case, fixed points may fall on the edges of the distribution, or even in areas without data. In this case, reconstruction error cannot be used as a good criterion. To address these issues, we introduce a density estimation process to probabilistically identify anomalies under these special circumstances. DE would assign a high likelihood to points in areas with dense data points and a low likelihood to others with low density. An anomaly is thus an input that is both far from its reconstruction (high reconstruction score) and statistically unlikely to have a high likelihood (low density estimation score). This synergy makes our method robust. As shown in Fig.2 and Eq.5, after forward reconstruction process (RE), we apply a stop-gradient operation $\text{sg}(\cdot)$ in learning of Gaussian distribution $\pi(z)$, which allows $\pi(z)$ to be learned independently without affecting the reconstruction training. In addition, the hyperparameter $\lambda$ in our total loss $\mathcal{L}_{\text{total}} = \mathcal{L}_{\text{rec}} + \lambda \mathcal{L}_{\text{MLE}}$ explicitly controls this trade-off. Our sensitivity analysis on $\lambda$ in Tab.5 shows that a balanced value ($\lambda = 0.5$) performs best, validating that the two objectives are complementary rather than purely conflicting. The reconstruction provides an error-based score, while the flow provides a likelihood-based score, and their combination is more powerful than either alone.

