# OpenReview forum: "Unifying Reconstruction and Density Estimation via Invertible Contraction Mapping in One-Class Classification"
_NeurIPS.cc/2025/Conference — NeurIPS 2025 poster_

### Official Review · Reviewer_Cw95 · 2025-06-06

**Clarity:** 2
**Significance:** 4
**Originality:** 3
**Rating:** 5
**Confidence:** 3

**Summary:**

This paper introduces an anomaly detection framework for one-class classification that combines reconstruction and density estimation through an invertible contraction mapping. The authors propose a reconstruction process where all inputs are iteratively mapped to a fixed point within the normal data manifold. This makes it so abnormal inputs yield high reconstruction errors. Additionally, the invertibility of the CM enables flow-based density estimation by learning a prior distribution over normal reconstructions and computing input likelihoods via the inverse mapping. By integrating both reconstruction-based and likelihood-based scoring, the method achieves strong performance across tabular and image datasets. Experiments and ablations demonstrate the value of both components in improving detection accuracy.

**Questions:**

While the theoretical analysis shows convergence of inputs toward fixed points within the normal manifold, it would be helpful to include a discussion of how this property behaves on real data, especially when dealing with ambiguous or near-boundary anomalies. Could this contraction mapping project mild anomalies too close to the manifold and reduce separability as a result?

**Ethical Concerns:**

["NO or VERY MINOR ethics concerns only"]

**Final Justification:**

The paper provides an interesting contribution to the field and the authors have addressed many of my and other reviewers' concerns so I recommend an accept.

**Limitations:**

The authors briefly acknowledge the limitation to tabular and image data, but the societal impact discussion is overly generic. It would strengthen the paper to include concrete considerations around false positives in safety-critical domains and the risk of over-relying on reconstruction-based signals in real-world deployments.

**Quality:**

3

**Strengths And Weaknesses:**

Strengths:

- The application of contraction mappings to enforce convergence of abnormal samples toward a fixed point on the normal manifold is theoretically grounded. The paper includes clear propositions and quantifies how reconstruction errors grow for anomalies.

- The proposed method merges two dominant OCC paradigms using a single CM-based architecture. This dual use of forward for reconstruction and reverse for flow-based likelihood estimation is efficient and clean.

- The approach outperforms state-of-the-art methods on tabular datasets (+3.78 AP and +0.75 AUROC vs MCM) and improves image-level and pixel-level AUROC on CIFAR-10 and MVTec AD.

- The authors conduct clear component ablations (reconstruction only, density only, full) and show sensitivity to hyperparameters to identify over-contraction issues when the product is too large.

Weaknesses:

- While the paper presents proofs around convergence of contraction mappings (e.g., fixed points within a normal manifold), it assumes the mapping generalises beyond the training data. There is no empirical validation of the fixed-point behavior on hard real-world anomalies.

- Iteratively mapping all inputs, including anomalies, toward a normal manifold may suppress outlier signals rather than enhance them. While this increases reconstruction error superficially, could it not also degrade interpretability or confuse downstream use cases? There should be a discussion on whether the model distinguishes subtle anomalies versus simply "normalising" everything.

- The CIFAR-10 experiments use a one-vs-all setting, which favours semantic class separation rather than true anomaly detection and does not reflect real-world conditions with subtle in-class deviations. While MVTec AD is a more appropriate benchmark, relying on a single real-world dataset limits the generality of the conclusions.

---

> ### Author Rebuttal · Authors · 2025-07-29
>
> ***Q1. Empirical validation of fixed-point behavior on hard, real-world anomalies. (Weakness 1 and Question 1)***
>
> >Thank you for your professional comments and insightful questions. The normal distribution of real world is irregular and complex, and we have also found these phenomena in experiments on tabular and image data. Although our Proposition 1 theoretically shows that fixed points will approach the normal manifold with optimization of reconstruction loss, real-world situations are even more complex. Following your comments, we discuss our reconstruction-only part on several specific situations and show corresponding empirical validation. In some simple cases, the reconstruction part is very effective, but in some complex or even extreme cases, the reconstruction part may fail.
>
> >**(i) Simple and general situations.** In actual experiments, we found that common normal manifolds are often connected and do not appear as ring-shaped or multi-ringed structures (see **Fig. 3(a)**). Under optimizatin of reconstruction loss, the fixed point is not completely located at the geometric center of the distribution, and sometimes , fixed points may even fall on the edges of the distribution. It is worth noting that in this simple case, the fixed points will all fall inside the normal manifold. Anomalies will are contracted towards the normal manifold, leading to more reconstruction error. In addition, some data manifolds are U-shaped (see **Fig. 3(d)**), the fixed point is located at the bottom of U. We found that after adjusting the random seed or training hyperparameters, the position of the fixed point will change and may not necessarily be located at the bottom position. But the same thing is that as long as enough training epochs are set, the fixed point is located inside or at the edge of U.
>
> >**(ii) Difficult and failure situations.** The normal data manifold in the real world is stochastic and complex, which may be non-connected (see **Fig. 4 (left)**), ring-shaped, or even multi-ring-shaped. In these complex cases, there is a high probability that the fixed point will be optimized to the non-data region and outside the manifold, and the fixed point may be very close to the abnormal data, resulting in false negative. In addition, if some abnormal data is very close to the normal manifold, almost all reconstruction-based methods will fail. To address these issues on reconstruction part, we introduce backward density estimation part to  learn probability distribution of original normal data. Even if the anomaly is close to normal in space, its likelihood may be much lower than that of normal data.
>
> ***Q2. Whether the mapping suppresses or enhances outlier signals. (Weakness 2)***
>
> >This is a very insightful observation. You are correct that our mapping 'normalizes' the abnormal features in the output space. However, this feature suppression is precisely the mechanism we leverage to amplify the anomaly signal in the form of reconstruction error. Next, we will show detailed reasons.
>
> >The key is that our reconstruction anomaly score is not the output $\mathbf{z}=\varphi(\mathbf{x})$ itself, but the discrepancy between the input and output, i.e., $||\mathbf{z}-\mathbf{x}||_{2}$. For a normal input $\mathbf{x}_n$, the contraction mapping acts nearly as an identity function, so $\mathbf{z}_n ≈ \mathbf{x}_n$ and the reconstruction error is minimal (see **lines 190-193**). For an abnormal input $\mathbf{x}_a$, the mapping pulls it towards the normal manifold, so $\mathbf{z}_a$ becomes significantly different from $\mathbf{x}_a$. This results in a large reconstruction error. Therefore, by suppressing the anomalous characteristics in the output $\mathbf{z}_a$, we create a large, easily detectable discrepancy signal. This enhances, rather than suppresses, the final anomaly score.
>
> ***Q3. Limitations of the experimental datasets. (Weakness 3)***
>
> >Thanks for your professional comments and valuable suggestions.
>
> >First of all, it should be noted that that we follow the initial works, DeepSVDD [1], and choose the one-vs-all setting for CIFAR-10 that is the classic setting in the OCC task. Using this established benchmark allows for direct and fair comparison against a wide range of state-of-the-art methods, which is essential for positioning our work. Next, we recognize the limitations of MVTec AD. Following your suggestions, we test our method on an extra non-toy industrial dataset, MPDD [2], with significant intra-class differences and a real-world medical datasets, ISIC [3].
>
> >**(i) MPDD.**  MPDD contains 1,346 images from 6 types of industrial metal products with varying lighting conditions, non-uniform backgrounds, and multiple products in each image. We continue to adopt the model and training settings (visual anomaly detection) in the manuscript for this experiment, where only resolution is adjusted to $256\times 256$. As shown in table below, compared to some classic methods, the performance of our method is competitive.
>
>
> |  | image-level AUROC | Pixel-level AUROC |Mean|
> | --- | --- | --- |  ---|
> | PaDim | 74.8 | 96.7 | 85.7 |
> | PatchCore | 82.1 | 95.7 | 88.9 |
> | CFlow | 86.1 | **97.7** | 91.9 |
> | Ours | **90.2** | 97.4 | **93.8** |
>
> >**(ii) ISIC.** ISIC is a real-word skin lesion dataset, originating from the ISIC 2018 challenge, contains dermoscopic images across seven categories. We employs 6,705 normal images from the official training set for model training and the remaining images, including normal and abnormal ones, for testing, where only classification label can be given and image-level AUROC,F1,ACC are envaluated.  We continue to adopt the same model and training settings as **(i)**. As a real-world dataset, defects are more diverse, and all models that perform well on the toy dataset perform poorly on this dataset. However, our method is superior compared to some classic methods.
>
>
> |  | AUROC | F1 | ACC |
> | --- | --- | --- | --- |
> | PatchCore | 68.09 | 61.98 | 59.26 |
> | FastFlow | 67.27 | 63.46 | 57.34 |
> | CFlow | 66.97 | 60.81 | 57.80 |
> | RD4AD | 76.48 | **67.94** | 67.72 |
> | SimpleNet | 69.01 | 60.66 | 56.99 |
> | Ours | **78.51** | 67.08 | **72.17** |
>
> >Finally, it should be noted that due to the limitation of rebuttal time, we did not fine tune more parameters to achieve optimal performance on two new datasets. In addition, the results on MPDD are sourced from [4], and the results on ISIC are sourced from [5].
>
> >[1] Lukas Ruff, et al. Deep one-class classification. In ICML, 2018.
>
> >[2] Stepan Jezek, et al. Deep learning-based defect detection of
> metal parts: evaluating current methods in complex conditions. In ICUMT workshop, 2021.
>
> >[3] Gutman David, et al. "Skin Lesion Analysis toward Melanoma Detection: A Challenge at ISBI 2016. In ISIC, 2016.
>
> >[4] Ximiao Zhang, et al. Realnet: A feature selection network with realistic synthetic anomaly for anomaly detection. In CVPR, 2024.
>
> >[5] Chunlei Li, et al. Scale-Aware Contrastive Reverse Distillation for Unsupervised Medical Anomaly Detection. In arXiv, 2025.
>
> ***Q4. Limitations and societal impact discussion. (Limitations)***
>
> >Thank you for this valuable suggestion. We also agree that a more concrete discussion of societal impact and practical risks is warranted. In fact, in our used tabular data, 'Arrhythmia' and 'Fraud' datasets are from real medical diagnostic data, where false positives may lead to some negative societal impact. Following your suggestions, we will expand the Limitations section (Appendix E) in the final version. We will add a discussion on the real-world consequences of detection errors (false positives/negatives) in safety-critical domains reflected in our datasets, such as medical misdiagnosis ('Arrhythmia') or failed fraud detection ('Fraud'). We will also connect this back to our method's design, arguing that our unified framework, which combines reconstruction with density estimation, provides a more robust signal than methods relying on a single paradigm, thereby potentially mitigating some of these risks in real-world deployments.

---

### Official Review · Reviewer_A2j8 · 2025-06-27

**Clarity:** 2
**Significance:** 3
**Originality:** 3
**Rating:** 5
**Confidence:** 3

**Summary:**

This paper proposes an unsupervised anomaly detection method that unifies reconstruction and density estimation via contraction mappings.
By employing Lipschitz-continuous mappings with a Lipschitz constant less than one, any input is guaranteed to converge to a unique fixed point.
Since this fixed point lies on the normal data manifold, reconstruction errors for anomalous inputs become significantly larger, enabling effective anomaly detection.
Moreover, the invertibility of the contraction mapping allows for density estimation in the style of flow-based models.
The proposed method jointly optimizes both reconstruction and density estimation through a bidirectional architecture,
achieving competitive or superior performance compared to existing methods on both tabular and image anomaly detection tasks.

**Questions:**

- If a contraction mapping is bijective, then its repeated application should also be bijective. However, since the mapping converges to a unique fixed point after repeated applications, it seems to contradict bijectivity. I understand that your method mitigates this issue by limiting the number of iterations. Still, increasing the number of iterations might make the reconstruction error for anomalies even larger. Could you elaborate on this trade-off and how it is handled in your method?

- While your method is based on reconstruction error, it is not encoder–decoder–based like an autoencoder. Instead, it uses a sequence of same-dimensional mappings. In such a case, there is a risk of the network collapsing into the identity mapping. Even with contraction mappings, if the Lipschitz constant is close to 1, the mapping could behave similarly to identity. How is this issue mitigated? My understanding is that the network design in Section 3.1.2 (Equation (2)) and the integration with a flow-based model help regularize the mapping by enforcing alignment with the prior. Is this correct?

- What is the motivation behind jointly optimizing reconstruction error and the flow-based likelihood? If \phi is trained to minimize reconstruction error, the transformed data distribution should resemble the original input distribution. On the other hand, maximizing the likelihood via a flow-based model pushes the transformed data to match a Gaussian prior. These two goals appear to be in conflict. Is my understanding correct, and if so, how is this trade-off addressed in your method?

- I also have some questions regarding the hyperparameters. For instance, setting $R=2$ or $3$ seems quite shallow compared to typical flow-based models, which often require deep compositions to capture complex distributions. However, from the ablation study, it appears that combining reconstruction and flow-based density estimation yields the best performance. Does this imply that the role of the flow in your method is not to model the data distribution in detail, but rather to act as a complementary scoring mechanism? I would appreciate it if you could clarify the intended role of the flow component.

If these concerns are resolved, I would be happy to increase both my overall score and confidence.

**Ethical Concerns:**

["NO or VERY MINOR ethics concerns only"]

**Final Justification:**

Incorporating contraction mappings into autoencoder-based anomaly detectors is an interesting idea, and the proposed method demonstrates strong empirical performance. In addition, the rebuttal has addressed my concerns. I recommend an Accept.

**Limitations:**

yes

**Paper Formatting Concerns:**

No major formatting issues observed.

**Quality:**

3

**Strengths And Weaknesses:**

## Strengths
- This paper is well-written and easy to follow.
- Models that minimize reconstruction error, such as autoencoders, can sometimes reconstruct both normal and anomalous data. The idea of using contraction mappings to address this issue is highly original and important. Additionally, leveraging the invertibility of contraction mappings to integrate with flow-based models is effective and appears to contribute meaningfully to anomaly detection performance.

## Weakness
- On the other hand, there are several points I did not fully understand or found somewhat unclear; I will list them in the “Questions” section.
- This is a very minor issue, but in line 183, I believe the expression should more precisely denote the L2 norm.

---

> ### Author Rebuttal · Authors · 2025-07-29
>
> ***Q1. The trade-off between convergence to a fixed point (iterations of $f$) and bijectivity (each $f$ unit). (Question 1)***
>
> >This is a very sharp and important observation. You are correct that infinite iterations of a contraction mapping unit $f$ would map all inputs to a single point, breaking bijectivity.  In the following response, we first need to clarify a mathematical conclusion that the composition of a finite number of invertible mappings is still an invertible mapping, but the composition of an infinite number of mappings is not necessarily an invertible mapping. Next, our method focuses on leverage a finite number of mapping composition to simulate iteration of a single contraction mapping, rather than directly using iteration of a single contraction mapping.
>
> >**(i) A general mathematical conclusion.**  First of all, we need to clarify that that the composition of a finite number of invertible mappings is still an invertible mapping, but the composition of an infinite number of mappings is not necessarily an invertible mapping. A simple counterexample: we consider that $f(x)=x/2$ is invertible with $x\in \mathbb{R}$ and has a unique inverse $f^{-1}(x)=2x$. the composition of an infinite number of $\lim_{n\to\infty} f^n(x) = \lim_{n\to\infty}x/2^n=0$. The infinite composite of this mapping is a constant function that is irreversible. So, in a finite number of composites, there is no contradiction between contraction mapping and reversibility.
>
> >**(ii) Consideration on mapping structure.**  According to **lines 166-168**, we apply $\varphi_{K}(\cdot)=f_{K}\circ f_{K-1}\circ \cdots f_{1}(\cdot)$ to simulate a finite number of iterations of $f$, rather than directly iterating $f$, where each $f_{k}$ has its own parameters $\theta_{k}$ and Lipschitz constant $\eta_{k}$. Lipschitz constant $\eta_{\varphi}\le \eta_{1}\cdot \eta_{2}\cdots \eta_{K}<1$ ensures contracting anomalies towards the normal distribution.  We leverage $\mathcal{L}_{\text{rec}}=||\varphi _ {K}(\mathbf{x})-\mathbf{x}|| _ {2}$ to train the entire network $\varphi _ {K}$, rather than firstly training a single $f$ and then iterating it a few times. From an optimization perspective, $\varphi _ {K}$ optimized by $\mathcal{L} _ {\text{rec}}$ tends to restore normal input to normal output instead of contracting all inputs (normal and abnormal) to one fixed point. From an mathematical perspective, a finite and small $K$ value gives $\varphi _ {K}$ the ability to contract the input to the normal manifold, but does not destroy the reversibility of $\varphi _ {K}$ .  In summary,  selecting $K$ as a finite value, the contraction properties of $\varphi _ {K}$ and the training objective jointly ensure that it can map anomalies to normal manifolds, but it will not completely shrink to a very small region or even a point and destroy its reversibility.
>
> >**(iii) Consideration on the number of mapping layer.**  In actual experiments, we intentionally keep $K$ and $R$ small to prevent over-contraction. This allows us to benefit from the "pull" of the contraction towards the normal manifold (which increases reconstruction error for anomalies) while does not destory the reversibility of our INN. The trade-off is managed by the choice of $K$ and $R$, which we study empirically in Table 5 (main manuscript). The results show that a moderate number of iterations yields the best performance (corresponding reasons also are provided in **lines 299-309**), validating our handling of this trade-off.
>
> ***Q2. The risk of collapsing to the identity mapping. (Question 2)***
>
> >The degradation towards identity mapping is a critical concern for contraction mapping trained with reconstruction loss. We show two explicit mechanisms to prevent this as follow, where one is the structure of contraction mapping unit and the other is regularity of our backward density estimation.
>
> >**(i)** According to **Eq.(2) and lines 197-198**, $f(x)=\left(\frac{\text{Id} + \text{G}}{2}\right)^{-1}(x) - x$, where $\text{G}$ is invertible and $\text{Id}$ is identity mapping. If $f$ degrades into an identity mapping under reconstruction loss, we have $\left(\frac{\text{Id} + \text{G}}{2}\right)^{-1}(x) - x = x$, which means $\left(\frac{\text{Id} + \text{G}}{2}\right)^{-1}(x) = 2x$. Based on definition of inverse mapping, we must have $\frac{\text{Id} + \text{G}}{2}(2x) = x$.  This means $\text{Id}(2x) + \text{G}(2x) = 2x$. Then, we can get $\text{G}(2x) = 0$ to make $f(x)$ an identity mapping. Paradoxically, $\text{G}$ is invertible and one-to-one mapping. Therefore, in terms of the structure of $f$, it is difficult to degrade into an identity mapping under the optimization of reconstruction loss.
>
> >**(ii)** The backward density estimation loss $\mathcal{L}_{\text{MLE}}$ provides a powerful regularizer against identity collapse. In the backward process, $\mathcal{L} _ {\text{MLE}}$ pushes $\varphi^{-1}$ to be a mapping that transforms a basic Gaussian distribution $\pi(\mathrm{z})$ to the (typically complex) input distribution $p(\mathrm{x})$. An identity mapping could only achieve this if $p(\mathrm{x})$ is also a Gaussian distribution, which is never the case. This forces $\varphi^{-1}$ to learn a non-trivial, meaningful transformation.
>
> ***Q3. The conflicting objectives of reconstruction and density estimation. (Question 3)***
>
> >Your understanding is perfectly correct. $\mathcal{L}_ {\text{rec}}$ pushes $\mathrm{z}$ to be close to $\mathrm{x}$, while $\mathcal{L}_ {\text{MLE}}$ pushes the distribution of $\mathrm{z}$ to be a Gaussian distribution. What we need to clarify is that both forward reconstruction (RE) and backward density estimation (DE) are indispensable, although they may seem contradictory. We have also designed corresponding trade-off schemes in the manuscript.
>
> >**(i) The motivation of integration of DE and RE.** When only using RE, if some anomalies are very close to the normal manifold or obtained fixed point in terms of spatial position, which causes small reconstruction errors and false negatives. In addition, the distribution of normal data may be very complex, chaotic, and disconnected (see **Fig 3&4**). In this case, fixed points may fall on the edges of the distribution, or even in areas without data. In this case, reconstruction error cannot be used as a good criterion. To address these issues, we introduce a density estimation process to probabilistically identify anomalies under these special circumstances. DE would assign a high likelihood to points on areas with dense data points and a low likelihood to others with low density. An anomaly is thus an input that is both far from its reconstruction (high reconstruction score) and statistically unlikely to have high likelihood (low density estimation score). This synergy makes our method robust.
>
> >**(ii) Our trade-off strategies.** As shown in **Fig.2 and lines 224-226**, after forward reconstruction process (RE), we apply a stop-gradient operation $\text{sg}(\cdot)$ in learning of Gaussian distribution $\pi(\mathrm{z})$, which allows $\pi(\mathrm{z})$ to be learned independently without affecting the reconstruction training.  In addition, the hyperparameter $\lambda$ in our total loss $\mathcal{L}_ \text{total} = \mathcal{L}_ \text{rec} + \lambda\mathcal{L}_ \text{MLE}$ (Eq.(7)) explicitly controls this trade-off. Our sensitivity analysis on $\lambda$ in **Table 3 (Appendix,Section D)** shows that a balanced value ($\lambda=0.5$) performs best, validating that the two objectives are complementary rather than purely conflicting. The reconstruction provides an error-based score, while the flow provides a likelihood-based score, and their combination is more powerful than either alone.
>
> ***Q4. Regarding the shallow depth of the flow network. (Question 4)***
>
> >Your understanding is correct. In fact, our goal is not to build a superior flow model compared to previous density-estimation-only methods and flow-based generative model, which would indeed require a much deeper architecture. According to our answer in **Q3.(i)**, our reconstruction-only method may may encounter some challenges caused by the spatial distribution of data. Therefore, the purpose of the flow component in our framework is precisely to act as an effective complementary scoring mechanism.
>
> >**(i)** The core of our method actually lies in the reconstruction part. A too large $R$ and $K$ will increase the network capacity, making reconstruction training difficult and requiring more training inference costs.
>
> >**(ii)** For fair comparision with previous tabular AD method, MCM [46], the value of $K \cdot R$ are roughly equivalent to the number of network layers of MCM (around six layers). In addition, visual AD method, CFlow [16], adopts the flow structure of 8 layers, while we set $K \cdot R = 9$ on MVTec AD dataset, which can actually meet the accuracy of visual anomaly detection.
>
> ***Q5. A minor issue of L2 norm notation. (Weakness 2)***
>
> >We thank the reviewer for their careful reading. We agree and will correct the notation $||\cdot||$ to $||\cdot||_{2}$ in line 183 and ensure consistency throughout the revised manuscript.

---

> > ### Comment · Reviewer_A2j8 · 2025-07-31
> >
> > Thank you for your response. My concerns have been addressed. I will raise my score to Accept.

---

> > > ### Author Response · Authors · 2025-08-01
> > >
> > > Thank you very much for your approval. Thanks again for your insightful comments and suggestions.

---

### Official Review · Reviewer_bUw6 · 2025-06-30

**Clarity:** 2
**Significance:** 3
**Originality:** 3
**Rating:** 4
**Confidence:** 3

**Summary:**

This paper proposes to resolve one class classification based unsupervised anomaly detection problem with thecontraction mapping. By iteratively transform samples through the contraction mapping, normal samples stayclose and abnormal samples moves far away, resulting large discrepency. The invertablity property of contractiohmapping enables backward density estimation, and thus present a unifed anomaly detection framework withforward reconstruction and backward density estimation. The method's efficacy is demonstrated throughextensive experiments on a variety of datasets and it shows competitive performance against currentstate-of-the-art methods.

**Questions:**

1. Are there any experiments to directly evaluate the effect of using contract mapping instead of indirect evaluation?

2. Can you further discuss the affects of different kinds of normal data distribution on CM results? It's interesting to see in what circumstances dose CM work well and in which case it may not well distinguish abnormal samples.

**Ethical Concerns:**

["NO or VERY MINOR ethics concerns only"]

**Final Justification:**

The author's response addressed my concerns. Referring to the reviews of other reviewers, I improved my score.

**Limitations:**

No Limitations

**Quality:**

3

**Strengths And Weaknesses:**

Strengths:

1.Introducing contract mapping into one class based anomaly detection is interesting2.The performance is promising3. The paper content and illustrations are clear and easy to follow

Weaknesses:
1. It seems that the main contribution largely comes from the introduced contraction mapping model. The author provides results for single forward reconstruction and backward density estimation in Table 3 and 4.in which the performance boost is observed. However, direct and clear evaluation of performance of contract mapping itself is needed. Just showing the RE an DE is indirect. From my perspective, they are just components of the unified framework.

2. Sensitivity of data distribution. Since the contraction mapping iteratively transforms data toward a fixed point derived by training the model on normal data, what if the normal data distribution itself is chaotic? Or if the normal data distribution is ring-shaped or U-shaped? Can the fixed point locate in an appropriate position? Will the reconstruction discrepancy identification work well in such cases?

As contraction mapping is an introduced mathematical tool, further justification for its use in anomaly detection should be supplemented.

---

> ### Author Rebuttal · Authors · 2025-07-29
>
> ***Q1. More direct and detailed evaluation on contraction mapping (CM). (Weakness 1 and Question 1)***
>
> >Thanks for your insightful comments and valuable suggestions. In fact, we compared the reconstruction-only CM (DE process is removed) with MLP-based and CNN-based Autoencoders in variants **(A)** and **(B)** in Tables **3&4**, but did not provide more detailed explanations for these experiments. Following your professional suggestions, we supplement these sections as follow. Moreover,we also add ablation studies of the components of reconstruction-only CM for your reference.
>
> >**(i)** In **Table 3 and lines 288-289**, to directly and fairly show the superiority of CM reconstruction, model **(A)** adopt the MLP structure with the same dimension of input and output, and the same number of layers for reconstruction as our CM. The significant performance improvement from Model **(A)** (AP/AUROC: 67.44/86.52) to Model **(B)** (75.21/90.32) represents a gain of +7.77 AP / +3.80 AUROC on all tabular datasets. This improvement can be directly attributed to the effectiveness of the contraction mapping in creating more discriminative reconstruction errors compared to a standard reconstruction approach.
>
> >**(ii)** As shown in **Table 4 (A&B)**, compared with the original CNN-based Autoencoder network, reconstruction-only CM variant **(B)** obtained gain of image-level AUROC by +4.29 on natural image data and image-level/pixel-level AUROC by +1.4/0.7. All of these directly demonstrate that the reconstruction of CM is superior on image datasets. In our manuscript, due to the space limitations, we did not provide a detailed comparison between variants **(A)** and **(B)** in the ablation experiment.
>
> >**(iii)** We directly provide a comparison between a non-CM invertible network and a CM invertible network. According to **Eq.(2)** in main manuscript **lines 197-198**, the core of each contraction mapping unit $f$ is the INN $\mathbf{G}$ that is a guarantee of contract mapping. Next, we test general INN network, Glow [1], widely used i-ResNet [2], and our used i-DenseNet [3] as $\mathbf{G}$ respectively in the following table. In these experiments, we only test reconstruction process. Although the structure of Glow is also an INN, each layer does not meet the requirements of CM ($\eta<1$), which makes $f$ no longer a CM. This also means that under the training of reconstruction loss, $f$ may degenerate into an identity map, without solving the 'well reconstructed anomalies' issue. On the contrary, i-ResNet and i-DensNet satisfy the conditions of our CM and achieve higher performance on used datasets. The comparison of the three groups also directly indicates that using CM for reconstruction is effective.
>
> ||All Tabular Datasets|CIFAR-10|MVTec AD|
> |---|---|---|---|
> |Glow [1]|66.14/84.39|71.27|97.3/96.5|
> |i-ResNet [2]|74.83/90.47|74.57|98.9/97.9|
> |i-DensNet [3]|75.21/90.32|76.88|99.2/97.9|
>
> >[1] Durk P Kingma, et al. Glow: Generative flow with invertible 1x1 convolutions. In NeurIPS, 2018.
>
> >[2] Jens Behrmann, et al. Invertible residual networks. In ICML, 2019.
>
> >[3] Yura Perugachi-Diaz, et al. Invertible densenets with concatenated lipswish. In NeurIPS, 2021.
>
> ***Q2. CM results on different kinds of normal data distributions. (Weakness 2 and Question 2)***
>
> >These questions you raised about the CM results on different normal data distributions are very crucial and very interesting. Sometimes, extreme and complex distributions may make reconstruction-only part ineffective, which is also the reason why we introduce density estimation process to collaborate with reconstruction process. Following your series of questions, we discussed several situations and provided some special cases of difficult reconstruction.
>
> >**(i) Ordinary circumstances.** In fact, the manuscript visualizes the position of the fixed point in different normal distributions. Generally, the fixed point is not completely located at the geometric center of the distribution, and sometimes due to different data or training settings, fixed points may even fall on the edges of the distribution (see **Fig.3 (a)**).
>
> >***Disconnected distribution.*** As shown in **Fig.4 (left)**, normal distribution is relatively scattered and chaotic, and the fixed point is located at the edge of one of the sub distributions. However, due to the fact that the abnormal distribution is far from the normal one, reconstruction error is still an effective criterion, achieving a good performance under reconstruction-only setting.
>
> >***U-shaped distribution.*** As shown in **Fig.3 (d)**, normal distribution is U-shaped and the fixed point is located at the bottom of U. We found that after adjusting the random seed or training hyperparameters, the position of the fixed point will change and may not necessarily be located at the bottom position. But the same thing is that as long as enough training epochs are set, the fixed point is located inside or at the edge of U. As long as the abnormal data is in a distant external area, CM will achieve good and stable performance (AP/AUROC: 88.98 $\pm$ 0.73 /97.18 $\pm$ 0.41 under reconstruction-only setting).
>
> >***Ring-shaped or multi-ring-shaped distribution.*** There is currently no typical ring-shaped normal distribution in the used datasets. **Fig.3 (b) and (c)** show multi-ring and irregular-ring structures, respectively. In fact, through multiple adjustments to the training hyperparameters and random seed, we found that fixed points may be located in some data free areas (central regions of the ring) or within the distribution (internal regions of the ring). If the anomaly is located in the central area of the ring, this method may theoretically fail, but this situation has not been found in experiments yet.
>
> >**(ii) Special difficult circumstances.** In the reconstruction-only experiments, we found that if the abnormal data is very close to the normal distribution in terms of spatial position, the reconstruction errors generated by CM for normal and abnormal data are very close, resulting in false negative and degrading performance. Of course, this is also a challenge for all reconstruction-based methods. In addition, In addition, due to the fact that fixed points may appear at any position on a normal manifold, they may be closer to some anomalies than other normal points, leading to false negatives.
>
> >Finally, in the most extreme case, when a few anomalies are within the normal distribution, our framework will fail. However, it should be noted that almost all existing OCC methods cannot address this type of problem. We believe this combined approach makes our method robust even for complex data distributions. We will add a discussion of this aspect to the limitations section in the appendix.
>
> >**(iii) Strategies for address difficult circumstances.** According to difficult circumstances in **(ii)**, CM reconstruction may fail due to the spatial position of fixed points and abnormal data. To address these issues, backward density estimation (DE) is introduced to learn probability distribution of original normal data. Even if the anomaly is close to normal in space, its likelihood may be much lower than that of normal data.
>
> >Regarding the typical ring-shaped situation where reconstruction with CM may fail, DE would assign a high likelihood to points on the ring but a low likelihood to points in the empty center. This synergy makes our method robust, as an anomaly would be flagged by either a large reconstruction error, a low likelihood, or both.

---

> > ### Author Response · Authors · 2025-08-08
> >
> > Dear Reviewer bUw6,
> >
> > I hope this message finds you well. As the discussion period is nearinits end with less than 2 days remaining, I wanted to ensure we have addressed all your concerns satisfactorily. If there are any additional points or feedback you'd like us to consider, please let us know.
> >
> > Thank you for your time and effort in reviewing our paper.

---

> > > ### Author Response · Authors · 2025-08-09
> > >
> > > We noticed that the discussion period is scheduled to conclude in just a few hours. We highly value your expert opinion and are very much looking forward to receiving your feedback. Even with the discussion period ending shortly, we are still eager to receive your comments and are available to address any concerns or questions you may have.

---

> ### Comment · Area_Chair_4344 · 2025-08-05
> **Please discuss rebuttal with authors**
>
> Please discuss rebuttal with authors

---

### Official Review · Reviewer_eeLX · 2025-07-02

**Clarity:** 2
**Significance:** 3
**Originality:** 2
**Rating:** 3
**Confidence:** 3

**Summary:**

The paper proposes a new method for unsupervised anomaly detection that combines a forward contraction mapping with density estimation via a reversed normalizing flow. The contraction mapping is designed to map any input data point toward a fixed point on the normal data manifold. Specifically in this paper, the mapping $f$ is chosen as a 1-Lipschitz neural network, regularized with spectral normalization and using the LipSwish activation function. The network is trained to minimize the reconstruction error $||f(x) -x||_2$ for arbitrary input
$x$. Then, leveraging the invertibility of $f$, the method then uses the inverse map of $f$ to transform a Gaussian  prior distribution into a real data distribution, allowing for estimating the likelihood of the input. Overall, the approach jointly minimizes the reconstruction error in the first stage and the negative log-likelihood in the second stage. The method is evaluated on multiple anomaly detection benchmarks and demonstrates competitive performance relative to existing anomaly detection methods.

**Questions:**

See Weaknesses

**Ethical Concerns:**

["NO or VERY MINOR ethics concerns only"]

**Final Justification:**

I maintain my score as "borderline reject"

**Limitations:**

See Weaknesses

**Paper Formatting Concerns:**

There's no concern on the format of the manuscript.

**Quality:**

3

**Strengths And Weaknesses:**

Strengths:
1. While the individual components of the proposed approach (i.e., the reconstruction map and the use of normalizing flows) draw heavily from prior work, the primary contribution lies in their combination. The experimental results provide reasonable evidence that combining the reconstruction map with normalizing flows leads to improved anomaly detection performance compared to using either component alone.
2. I appreciate that the authors included a detailed sensitivity analysis of the hyperparameters $\lambda$ and $\alpha$, which  illustrate the trade-off between reconstruction error and the MLE loss.

Weaknesses:
1. The term "well-constructed anomalies" was not defined or explained in the introduction, which makes it difficult to grasp the motivation behind the proposed method. It appears to refer to reconstructed anomalous inputs that fall outside the normal data manifold, and I am not even sure if I understand this correctly. Clarifying this early on would make the paper more accessible, especially for
    readers less familiar with reconstruction-based anomaly detection.
2. The theoretical guarantees provided for the reconstruction map appear shallow. In particular, Proposition 1 does not rigorously prove that the fixed point $x^*$ will be close to the original input $x$. The paper does not show that, for any chosen mapping $f$ and optimization algorithm, the reconstruction loss $||f(x) - x||_2$ can be made arbitrarily small.
3. The choice of the activation function $\psi$ of the neural network employed appears important to me, yet it is omitted from the main text. Only in the appendix do the authors mention that $\psi$ is the  LipSwish activation function.  Given that ReLU is also 1-Lipschitz and commonly used, why do the author choose LipSwish over ReLU? Does LipSwish bring empirical benefit, possibly related to the extra learnable parameter $\beta$?
4.   The paper lacks a theoretical justification for why the integration of reconstruction mapping and normalizing flows should be more effective than using either component independently.

---

> ### Author Rebuttal · Authors · 2025-07-29
>
> ***Q1. Vague term makes it difficult to understand the core motivation of the paper. (Weakness 1)***
>
> >We thank you very much for pointing out the ambiguity of this term.
>
> >**(i)** Generally, reconstruction-based methods rely on an assumption: reconstruction network trained on normal samples cannot reconstruct abnormal samples. Ideally, the trained network cannot reconstruct an abnormal sample well during inference, so there is a significant difference between the abnormal input and its reconstruction result. However, general reconstruction network (Autoencoder) often fails to achieve the ideal situation mentioned above and **still reconstructs some abnormal inputs well**. Thus, "well-reconstructed anomalies" refers to a common failure mode in reconstruction-based anomaly detection methods. To make this clearer for all readers, we will explicitly define the term "well-reconstructed anomalies" in the introduction of the revised manuscript and connect it directly to the problem of low reconstruction error for abnormal samples.
>
> >**(ii)** Our core motivation of reconstruction via contraction mapping is to combat the above failure mode. Instead of allowing the network to freely reconstruct inputs, we constrain the output to lie on the normal manifold. As shown empirically in **Fig. 1(d)** and **Fig. 4**, our contraction mapping iteratively pulls abnormal inputs towards the normal manifold. This process intentionally increases the reconstruction error for anomalies, making them more detectable and directly addressing the "well-reconstructed anomalies" problem. We will explicitly state this in the introduction of the revised manuscript.
>
> ***Q2. Explanation of Proposition 1. (Weakness 2)***
>
> >Thanks for your insightful comments on the depth of our theoretical analysis.
>
> >**(i)** Proposition 1 mainly focuses on establishing a direct causal link between our optimization and the fixed point's location. The proposition (**line 151-159**) states that $||\mathrm{x}^* -\mathrm{x}|| _ {2}\le \mathcal{L}_{\text{rec}}/(1-\eta )$, which directly and rigorously demonstrates that by minimizing the reconstruction loss $\mathcal{L} _ {\text{rec}}$ on normal samples $\mathrm{x}$ (our training objective), we are guaranteed to bound the distance to the fixed point $\mathrm{x}^*$. The empirical visualizations in **Fig. 3** on real datasets strongly support this theoretical claim, showing the fixed point landing within the normal data manifold.
>
> >**(ii)** We are unable to prove that for any mapping and optimization algorithm, $\mathcal{L}_{\text{rec}}$ can be made arbitrarily small, because this depends on model capacity and data distribution complexity. This is a fundamental challenge in deep learning optimization. The empirical visualizations in Fig. 3 on real datasets, which show the fixed point landing within the normal data manifold, provide strong support for this theoretical claim. In addition, as shown in **line 153**, we only use the term 'a small neighborhood' rather than 'arbitrarily small' or 'complete convergence'.
>
> ***Q3. Why apply LipSwish activation function. (Weakness 3)***
>
> >Thanks for your professional comments. We note that the choice of LipSwish is deliberate and critical. LipSwish guarantees the reversibility of the entire network, while ReLU cannot guarantee it. We provide a specific analysis below for your reference.
>
> >**(i)** In this work, we apply an invertible network $\varphi$ as the foundation, where LipSwish activation function is invertible and ensures the invertibility of $\varphi$. In contrast, $\mathbf{ReLU}(x)=\mathbf{Max}(0,x)$ maps all negative inputs to zero, which creates a many-to-one mapping that is mathematically non-invertible. When $x < 0$, the input information is permanently lost, making it impossible to recover the input from the output.
>
> >**(ii)** As mentioned in the appendix (**line 45,appendix**), we adopt i-DenseNet [1] to parameterize our mapping $\mathbf{G}$ in **Eq.(2)**. According to [1], the use of LipSwish is a standard and effective choice in this context, providing more expressive power than simpler activations. LipSwish is also smooth, continuously differentiable, and invertible, which are essential properties for stable training of INNs.
>
> >[1] Yura Perugachi-Diaz, et al. Invertible densenets with concatenated lipswish. In NeurIPS, 2021.
>
> ***Q4. Why the integration of reconstruction and density estimation should be more effective. (Weakness 4)***
>
> >The two components are not just used together and they are synergistic.
>
> >**(i)** When only using reconstruction mapping, we found that some anomalies were very close to the normal manifold in terms of spatial position, which caused small reconstruction errors and false negatives. In addition, the distribution of normal data may be very complex, chaotic, and disconnected. In this case, fixed points may fall on the edges of the distribution, or even in areas without data. In this case, reconstruction error cannot be used as a good criterion. To cope with the above situations, we introduce a density estimation process to probabilistically identify anomalies near the normal manifold.
>
> >**(ii)** Generally, an anomaly is thus an input that is both far from its reconstruction (high reconstruction score) and statistically unlikely to have high likelihood (low density estimation score). Therefore, we apply a integration of reconstruction and density estimation to identify test input (see **Eq.(8)**).
>
> >**(iii)** Our ablation studies (**Table 3 & 4**) provides strong support for this synergy. The full model **(D)** consistently and significantly outperforms the reconstruction-only **(B)** and density-only **(C)** variants, demonstrating that their combination is more powerful than either part alone.

---

> > ### Author Response · Authors · 2025-08-08
> >
> > Dear Reviewer eeLX,
> > ﻿
> > I hope this message finds you well. As the discussion period is nearinits end with less than 2 days remaining, I wanted to ensure we have addressed all your concerns satisfactorily. If there are any additional points or feedback you'd like us to consider, please let us know.
> > ﻿
> > Thank you for your time and effort in reviewing our paper.

---

> ### Comment · Area_Chair_4344 · 2025-08-05
> **Please discuss rebuttal with authors**
>
> Please discuss rebuttal with authors

---

### Note · Authors · 2025-08-13

Dear Area Chairs, Senior Area Chairs, and Program Chairs,

We thank all the reviewers for their time and effort in providing constructive feedback on our work. We would like to summarize the review process and re-emphasize our core contributions for your final consideration.

Our work is the first to introduce contraction mapping (CM) to the field of anomaly detection. Based on the theory of CM, we propose a new CM-based AD theory related to reconstruction-based methods and provide both theoretical and experimental evidence. In addition, we unify the two dominant paradigms of AD (reconstruction-based and density-based methods) by applying the invertible contraction mapping. The synergy between forward reconstruction and backward density estimation is carefully designed and validated through both deep mathematical considerations and extensive experiments instead of simple ensemble (**see rebuttal to Reviewer A2j8**). Our method demonstrates excellent generalization, achieving competitive performance across diverse domains, including tabular, natural image, industrial, and medical data (**see rebuttal to Reviewer Cw95**).

After a more detailed analysis and explanation of our method, **Reviewer A2j8** raised the original score to Accept. After providing the supplement of the method generalization, **Reviewer Cw95** also recommended an Accept. Although **Reviewer eeLX** maintained a 'borderline reject' and did not provide any further reasons, we consider that it may be because we did not provide proof for the proposition they proposed, i.e., ***for any chosen mapping $f$ and optimization algorithm, the reconstruction loss can be made arbitrarily small***. It should be noted that this proposition seems to lack some specific conditions and assumptions, making it very difficult to directly prove. We also explain the reasons in the corresponding rebuttal. **Reviewer bUw6** seems to have missed the discussion stage, but we still appreciate their insightful initial comments and valuable suggestions. If they still have concerns after reading our rebuttal, we hope they can refer to the rebuttal we provided to other reviewers.

Our paper presents a novel and effective approach to anomaly detection under OCC setting, backed by a combination of theory and experiment. We hope that this unified framework can contribute to the anomaly detection community.

Thank you for your time and careful consideration.

---

### Decision · Program_Chairs · 2025-09-17

**Decision:**

Accept (poster)

**Comment:**

The paper proposes a new framework for one-class classification by introducing invertible contraction mappings to unify reconstruction and density estimation. The authors effectively addressed reviewers' concerns regarding theoretical depth, validation, and clarity, with rebuttals supported by additional experiments and analysis. While one reviewer maintained a borderline reject, this reviewer did not provide any specific reasons after reading the author response. Therefore, I recommend accepting the paper.